# Chromosome length is constrained by spindle scaling to ensure faithful mitosis in mammals

Yu-Long Zhao [1,2,3,4,9], Yi-Ming Zhao [5,9], Yi-Fang Yan [6,9], Ning Yang [1,2,3,4,9], Si-Nan Ma [1,2,3,4,9], Rui-Jia Wang[5], Gui-Hai Feng [1,2,3,4✉], Zhi-Kun Li [1,2,3,4,7✉], Wei Li [1,2,3,4✉] & Li-Bin Wang [5,6,8✉]

## Abstract

**Why eukaryotic genomes are universally divided among multiple chromosomes remains an unresolved question. Although yeast and mouse cells can tolerate chromosomal fusions without impairing viability, we show here that chromosome length in mammalian cells is constrained by a biophysical limit governed by spindle geometry. Using engineered mouse cells carrying fused chromosomes of defined sizes, we identify ~308 Mb as the maximal length tolerated for faithful mitosis. Chromosomes exceeding this threshold disrupt segregation, leading to daughter cell re-coalescence and polyploidization. Aurora B kinase regulates this process by modulating spindle elongation; its inhibition induces mitotic failure even in chromosome configurations within the tolerated threshold of ~308 Mb. These findings explain the structural basis for genome fragmentation in animals and reveal a general mechanism linking chromosome size, spindle dynamics, and genome stability.**

**Keywords** Mitosis; Chromosomal Translocations; Spindle; Chromosome Length; Polyploidization
**Subject Categories** Biotechnology & Synthetic Biology; Cell Cycle; DNA Replication, Recombination & Repair

## Introduction

Genome structure varies greatly across biological kingdoms, yet a consistent feature distinguishes eukaryotes: the organization of their genomes into multiple linear chromosomes (Bendich and Drlica, 2000; Coghlan et al, 2005; Krawiec and Riley, 1990). In contrast, viruses and prokaryotes typically rely on compact, often single-molecule genomes—either linear or circular—that replicate and segregate without the need for chromatin or centromeric scaffolds (Ghosh et al, 2006). While eukaryotic chromosomal partitioning enables higher-order regulation, its universality remains puzzling. Notably, synthetic yeast with a single fused chromosome remains viable, suggesting that genome segmentation is not intrinsically required for cell survival or gene regulation (Gu et al, 2022; Shao et al, 2018; Shao et al, 2019).

This observation raises a key question: what constraints prevent natural eukaryotes from consolidating their genomes into fewer chromosomes? A possible answer comes from plants, where mitotic failure occurs when chromosome arms exceed half the spindle axis length, pointing to a geometric threshold for chromosome segregation (Hudakova et al, 2002; Schubert and Oud, 1997). However, it is unknown whether such physical limits apply to animal cells, which lack the spatial restrictions imposed by plant cell walls (Schubert and Vu, 2016).

Here, we demonstrate that mammalian cells exhibit a similar spindle-dependent threshold for chromosome segregation. Using engineered mouse stem cells with defined chromosomal fusions, we show that chromosomes longer than the spindle axis trigger polyploidization through failed mitosis. Aurora B kinase adjusts spindle length to accommodate enlarged chromosomes, but its inhibition collapses this adaptation, promoting segregation failure. These findings uncover a previously underappreciated physical limit on chromosome size in animals, offering a mechanistic rationale for the evolutionarily conserved fragmentation of eukaryotic genomes.

## Results

### Chromosomal translocations disrupt mitotic dynamics and trigger ploidy changes

To systematically investigate how chromosomal translocations influence ploidy transitions, we employed our previously established head-to-tail chromosome ligation strategy in haploid embryonic stem cells (haESCs) (Wang et al, 2022). We selected haESCs for this study because they contain a single set of chromosomes, which facilitates precise genome engineering.

[1]Key Laboratory of Organ Regeneration and Reconstruction, State Key Laboratory of Stem Cell and Reproductive Biology, Institute of Zoology, Chinese Academy of Sciences, 100101 Beijing, China. [2]Institute for Stem Cell and Regeneration, Chinese Academy of Sciences, 100101 Beijing, China. [3]University of Chinese Academy of Sciences, 100049 Beijing, China. [4]Beijing Institute for Stem Cell and Regenerative Medicine, 100101 Beijing, China. [5]Department of Cell Biology, School of Basic Medical Sciences, Peking University Stem Cell Research Center, Peking University Health Science Center, Peking University, 100191 Beijing, China. [6]State Key Laboratory of Female Fertility Promotion, Department of Obstetrics and Gynecology, Peking University Third Hospital, 100191 Beijing, China. [7]School of Life Sciences, Sun Yat-sen University, 510275 Guangzhou, China. [8]NHC Key Laboratory of Birth Defects Prevention, 451163 Zhengzhou, China. [9]These authors contributed equally: Yu-Long Zhao, Yi-Ming Zhao, Yi-Fang Yan, Ning Yang, Si-Nan Ma. ✉E-mail: fenggh@ioz.ac.cn; lizhk58@mail.sysu.edu.cn; liwei@ioz.ac.cn; wanglibin@bjmu.edu.cn

Specific chromosome pairs—Chr2 + 1, Chr2 + 3, and Chr4 + 5—were fused in a defined orientation (Figs. 1A and EV1). As a control, we included a previously characterized mis-translocated line, Chr1 + 2′, whose chromosome length is intermediate between the wild-type (WT) and Chr4 + 5 (Wang et al, 2022). Notably, haESCs are known to spontaneously diploidize during culture and give rise to stable diploid ESCs (diESCs), which typically maintain a constant ploidy over extended passages (Elling et al, 2011; Leeb and Wutz, 2011). This intrinsic stability makes them an ideal system to monitor ploidy alterations, if any, resulting from chromosome engineering. Strikingly, diESCs derived from Chr2 + 3 and Chr2 + 1 lines underwent progressive polyploidization over 26 days of culture, whereas Chr1 + 2′ and Chr4 + 5 lines retained diploid stability comparable to WT controls (Fig. 1B).

To investigate mitotic regulation in polyploidization, we conducted cell cycle analyses comparing WT with chromosome-translocated diESCs. Interestingly, cell cycle profiling uncovered a pan-translocation effect: all rearranged diESCs showed an increase in the proportion of cells in G2/M phase (Fig. 1C), suggesting these types of translocations broadly delay mitotic progression. To determine if consistent transcriptional alterations underlie mitotic progression delays in chromosome-translocated cells, we performed transcriptomic analysis (Data ref: Wang et al, 2022). We identified 67 consistently dysregulated genes across all translocated cell lines (Fig. 1D). However, pathway enrichment analysis of these genes revealed no direct association with cell cycle checkpoints (Fig. 1E). Furthermore, we did not observe consistent enrichment patterns across the three individual cell lines (Fig. EV2). Collectively, these results suggest that the observed mitotic stress is likely induced by physical constraints rather than consistent transcriptomic changes.

## Spindle–chromosome scaling determines segregation fidelity

Previous studies demonstrate that centromere dissociation-induced lagging chromosomes prolong anaphase duration to enable lagging chromosome reintegration (Afonso et al, 2014). To systematically investigate the mitotic integration dynamics of lagging chromosomes induced by chromosomal translocations during anaphase, we established H2B-RFP-labeled diESCs for real-time chromosomal visualization. Through three-dimensional imaging analysis, we observed that translocated chromosomes exhibited persistent lagging behavior throughout anaphase progression (Movies EV1–5). To ascertain whether lagging chromosomes are fused chromosomes, we used whole chromosome FISH to label fused chromosomes and found that the lagging chromosomes were perfectly merged with fish probes (Fig. EV3A). Notably, Chr2 + 3 and Chr2 + 1 diESC lines demonstrated complete failure of chromosomal segregation at anaphase termination (Fig. 2A; Movies EV4 and EV5). To quantitatively assess the segregation competence of these lagging chromosomes, we conducted longitudinal live-cell imaging analyses. Time-lapse imaging showed that chromatin bridges at anaphase occurred in 6.1% (8/131) of Chr4 + 5 diESCs, 82.1% (64/78) of Chr2 + 3 diESCs, and 83.9% (94/112) of Chr2 + 1 diESCs. However, the presence of a chromatin bridge did not necessarily result in cell co-coalescence: co-coalescence was observed in Chr2 + 3 and Chr2 + 1 diESCs but not in Chr4 + 5 diESCs (Fig. 2B; Movies EV6–10). Previous studies have shown that

chromatin bridges during telomere crisis can result in binucleated or multinucleated cells (Davoli and de Lange, 2012). Consistent with these findings, we frequently observed binucleated cells during the re-coalescence of Chr2 + 3 and Chr2 + 1 diESCs (Fig. 2B; Movies EV9 and EV10). Extending these previous reports, we found that chromatin bridges persisted not only in binucleated cells but also in tri-nucleated cells (Fig. EV3B). Taken together, our results suggest that binucleation likely represents an intermediate stage in the process of cell re-coalescence. To elucidate the potential mechanical constraints underlying this phenomenon, we performed comparative morphometric analyses between translocated chromosome length and spindle architecture. The lengths of the longest chromosomes and the spindle axes were measured at the end of anaphase (Fig. 2C). Quantitative measurements demonstrated a significant extension of spindle axes in all chromosome-translocated diESC variants compared to WT counterparts (Fig. 2D). Strikingly, in both Chr2 + 3 and Chr2 + 1 diESC lines, the physical dimensions of translocated chromosomes exceeded the average half-spindle axes length (Figs. 2E and EV3C). This observation suggests that surpassing this critical geometric threshold may disrupt the force equilibrium required for faithful segregation, thereby predisposing to chromosome re-fusion and subsequent polyploidization through failed cytokinesis.

Tetraploid cells typically have larger volumes, which theoretically accommodates enlarged mitotic spindles. To test this, we quantified spindles in tetraploid ESCs from WT, Chr2 + 3, and Chr2 + 1 lines, with haploid and diploid WT ESCs as controls. As expected, spindle axes length increased with ploidy. Moreover, tetraploid Chr2 + 3 and Chr2 + 1 ESCs exhibited even longer spindle axes than WT tetraploids (Fig. 2F). In these two lines, the average half-spindle axes length significantly exceeded that of the longest chromosomes (Figs. 2F and EV3C). Consistent with this adaptation, both Chr2 + 3 and Chr2 + 1 tetraploid ESCs maintained stable spindle dimensions over extended culture and achieved successful chromosome segregation in all observed anaphase events (102 and 105 events, respectively), despite frequent lagging chromosomes (Figs. 2G and EV3D). In addition, fluorescence-activated cell sorting (FACS) analysis of long-term cultured tetraploid Chr2 + 3 and Chr2 + 1 ESCs confirmed that, as a population, these chromosome-translocated tetraploid cells maintained robust segregation capacity (Fig. 2G). These results highlight the adaptive advantage of tetraploidy in accommodating oversized chromosomes through spindle scaling mechanisms.

## Aurora B kinase regulates spindle adaptation

Physically, the persistent presence of lagging chromosomes within the central spindle region imposes a spatial constraint during mitosis—specifically, it can obstruct the physical ingression of the contractile ring and thereby impede successful cytokinesis (Lacroix and Maddox, 2012). However, this geometric interference alone does not fully explain the subsequent cellular response, particularly the spindle elongation we observed and the eventual daughter cell re-coalescence that leads to genome doubling.

To investigate the mechanistic link between persistent lagging chromosomes and spindle elongation, we quantified the localization of Aurora B kinase. This key component of the chromosomal passenger complex monitors chromosome segregation and prevents premature de-condensation during anaphase (Afonso et al, 2014).

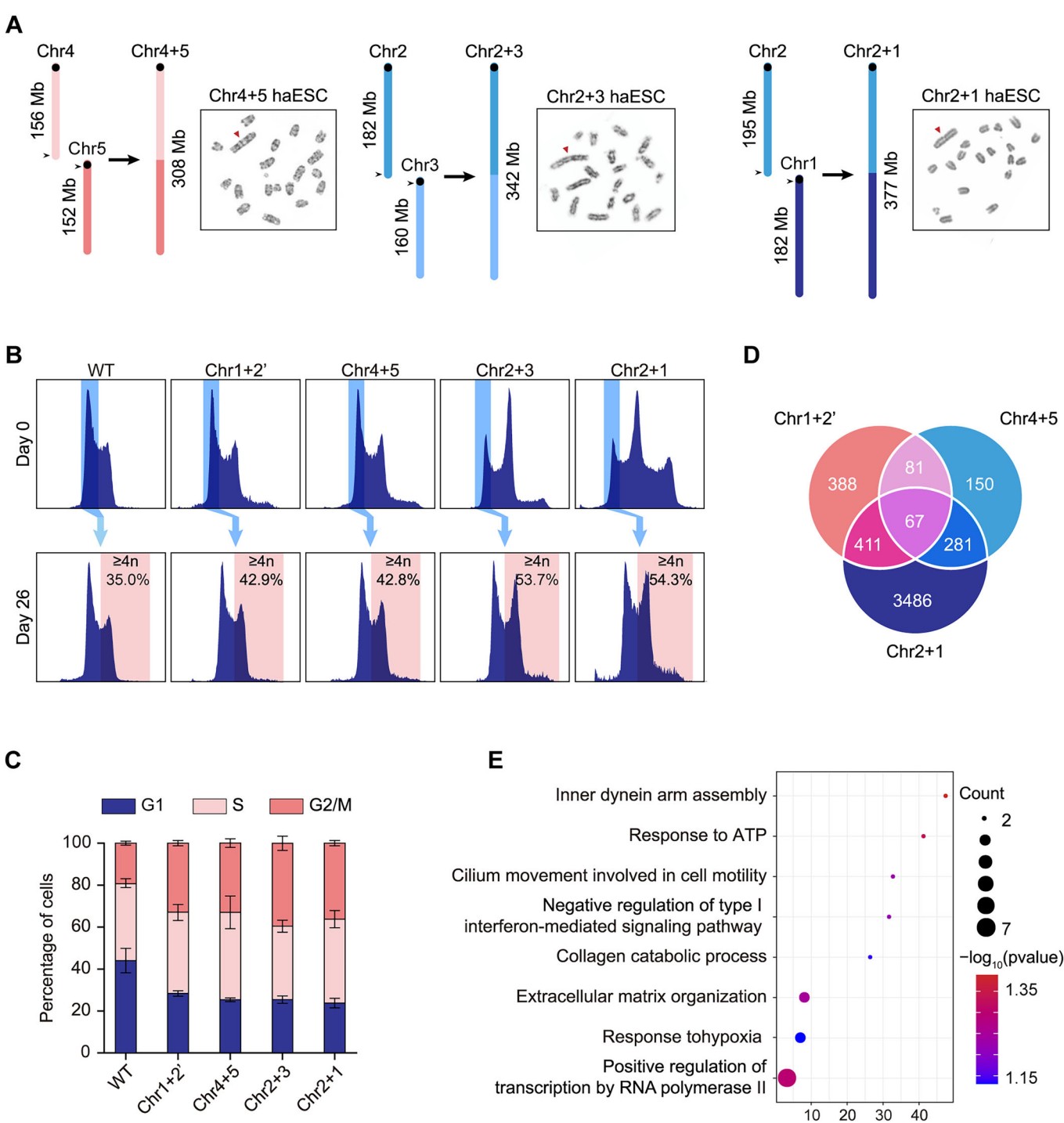

**Figure 1.   Mitotic defects and ploidy transitions induced by chromosome translocations in diploid ESCs.**

(A) Schematic diagrams of reciprocal translocations between chromosomes 4 and 5 (Chr4 + 5), 2 and 3 (Chr2 + 3), and 2 and 1 (Chr2 + 1), with representative G-banded karyotypes. Translocated chromosomes are indicated by red triangles. (B) Ploidy dynamics in WT and translocation lines. Top: Flow cytometry gating strategy for 2n cell populations. Bottom: Quantification of polyploid cells (≥ 4n) at Day 26. The x axis indicates DNA content quantified by Hoechst fluorescence intensity, and the y axis represents cell count. (C) Cell cycle analysis of WT and chromosome-translocated diESCs. Each bar represents the cell cycle of three biological replicates (n = 3). Data are presented as mean ± SD. (D) Venn diagram of differentially expressed genes (DEGs) shared among Chr1 + 2', Chr4 + 5, and Chr2 + 1 lines. (E) Gene Ontology (GO) enrichment analysis of 67 conserved DEGs across all translocation lines. Enrichment analysis was calculated by using Fisher's exact test. Source data are available online for this figure.

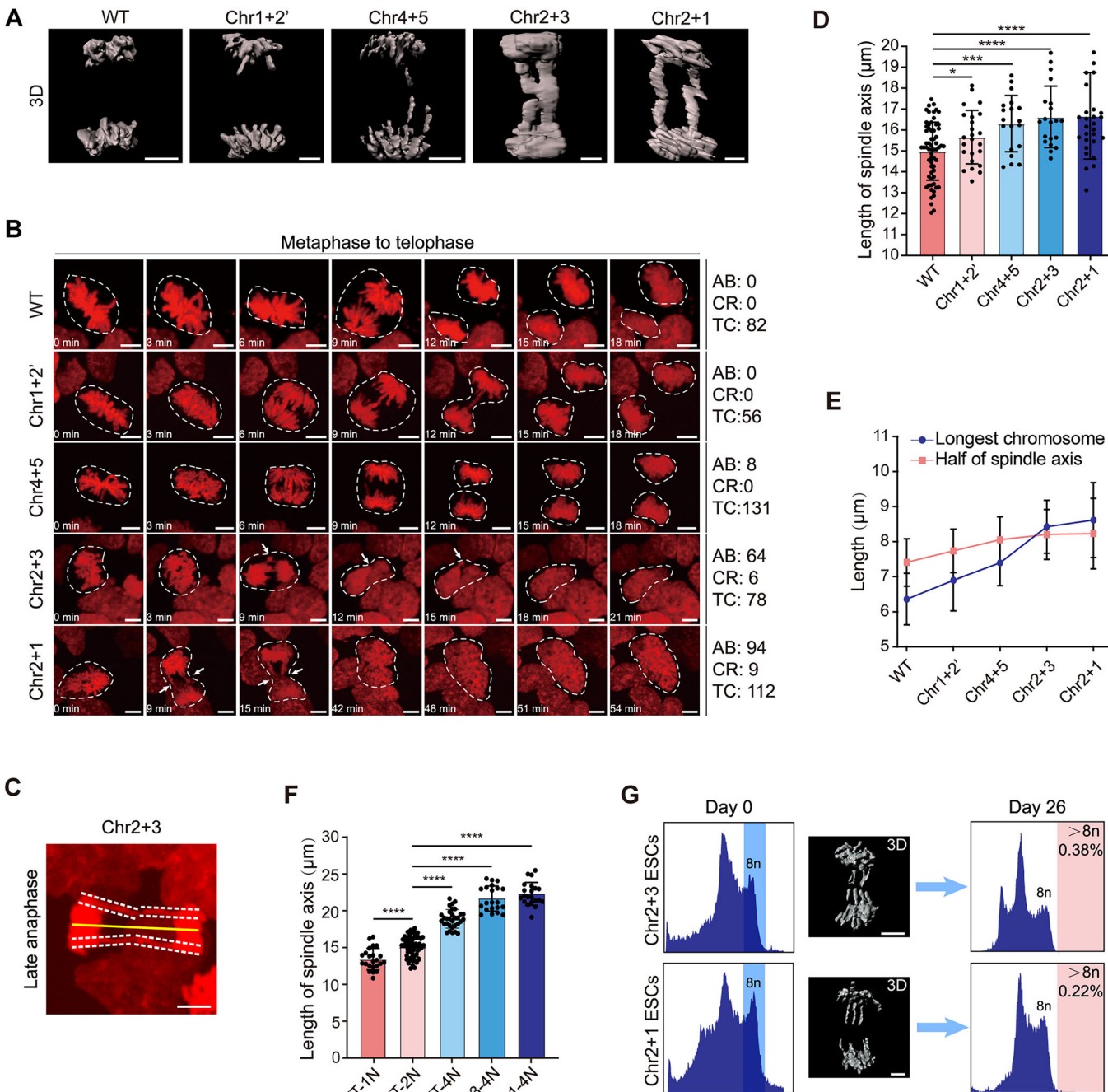

**Figure 2. Altered spindle–chromosome scaling drives segregation errors in translocated ESCs.**

(A) 3D-reconstructed anaphase chromosomes in WT and translocation lines. Scale bar = 5 μm. (B) Time-lapse imaging of metaphase-to-anaphase progression (0 min = metaphase plate formation). White dotted circles indicate cells undergoing mitosis, and white arrows mark lagging chromosomes. Scale bar = 5 μm. For each group, numbers for AB, CR, and TC represent pooled cell counts from multiple independent live-cell imaging experiments, including cells with complete mitotic progression. (C) Measurements of the spindle axis and the longest chromosomes. The yellow solid line indicates the spindle axis, and white dotted lines indicate the longest chromosomes. Scale bar = 5 μm. (D) Spindle axis length comparison during anaphase. WT: $n = 63$; Chr1 + 2′: $n = 32$; Chr4 + 5: $n = 19$; Chr2 + 3: $n = 20$; Chr2 + 1: $n = 29$. Statistical analyses were performed using unpaired $t$ test. Data are presented as mean ± SD, with *$P$ value of 0.0417, ***$P$ value of 0.0004, and ****$P$ value of 7.8e-05 and 1.76e-04 rightward correspondingly. (E) Correlation between the longest chromosome length and half of the spindle axis length in WT, Chr1 + 2′, Chr4 + 5, Chr2 + 3, and Chr2 + 1 diESCs. Half of the spindle axis length data are from (D), and chromosome length data are from Fig. EV3C. Data are presented as mean ± SD. (F) Spindle axis length comparison between haploid, diploid, and tetraploid cells. WT-1N: $n = 23$; WT-2N: $n = 63$; WT-4N: $n = 31$; 2 + 3-4 N: $n = 21$; 2 + 1-4 N: $n = 21$. Each data point represents the measured length of the spindle axis. Statistical analyses were performed using unpaired $t$ test. Data are presented as mean ± SD, with ****$P$ value of 1.2e-05, 2.4e-14, 8.52e-12, and 8.53e-12 rightward correspondingly. (G) 3D images and FACS analyses of tetraploid Chr2 + 3/2 + 1 ESCs. Blue rectangle: sorting of cells with DNA content equal to 8n. Pink rectangle: analyses of cells with DNA content >8n. Scale bar = 5 μm. Source data are available online for this figure.

Immunofluorescence quantification of Aurora B in late anaphase cells revealed the following incidences of chromosome bridges: WT (0/10), Chr1 + 2' (0/10), Chr4 + 5 (8/15), Chr2 + 3 (8/8), and Chr2 + 1 (8/8) diESCs. Notably, Chr4 + 5 diESCs with bridges, as well as Chr2 + 3 and Chr2 + 1 diESCs, exhibited heightened Aurora B accumulation at these sites (Fig. 3A,B). This suggests that Aurora B may coordinate spatial adaptations in spindle architecture to ensure chromosomal resolution.

This observation prompted us to directly examine the functional role of Aurora B in regulating the segregation of translocated chromosomes. We pharmacologically inhibited Aurora B using ZM447439—a selective Aurora B kinase inhibitor (Fig. EV4A)—in both WT and chromosome-translocated diESCs (Chr4 + 5 and Chr2 + 1). Strikingly, Aurora B inhibition significantly attenuated spindle elongation across all genotypes (Fig. 3C), while having no detectable effect on chromosome length (Fig. EV4B).

Quantitative morphometric analyses revealed a critical mechanical mismatch: in ZM447439-treated Chr4 + 5 diESCs, the translocated chromosomes exceeded half the spindle axis length (Fig. 3D), a marked departure from the coordinated chromosome–spindle scaling observed under untreated conditions (Fig. 2E). This spatial incompatibility correlated with pronounced mitotic failure, resulting in polyploidization in 14.4% of Chr4 + 5 diESCs (15/104), closely resembling the intrinsic segregation defect seen in Chr2 + 1 diESCs regardless of Aurora B activity (Figs. 3E and 2E; Movies EV11–13). FACS profiling over 18 days confirmed robust progressive polyploidization in Chr4 + 5 diESCs (Fig. 3F). Aurora B inhibition markedly accelerated polyploidization in Chr2 + 1 diESCs, while resulting in only a minor increase in polyploidy in WT diESCs—a finding consistent with the previous report (Figs. 3F and EV4C) (Hindriksen et al, 2015).

Together, these findings demonstrate that Aurora B activity is required to maintain spindle–chromosome scaling under conditions of increased chromosomal length. Inhibition of Aurora B disrupts this adaptation, leading to segregation failure and polyploidization even in Chr4 + 5 cells, which otherwise maintain mitotic fidelity. These results suggest that Chr4 + 5 lies close to the upper boundary of chromosome length that can be faithfully segregated in diploid cells under physiological spindle dynamics.

### In vivo validation of translocation-induced polyploidization

To determine whether the relationship between chromosome length and mitotic fidelity observed in mouse embryonic stem cells is specific to this cell type or extends to broader biological contexts, we investigated whether polyploidization also occurs in vivo during differentiation. To this end, we utilized the pluripotency of diESCs in a teratoma formation model. Diploid ESCs carrying various chromosomal configurations (WT, Chr4 + 5, Chr2 + 3, and Chr2 + 1) were subcutaneously injected into severe combined immunodeficiency (SCID) mice. After 4 weeks, all cell lines formed teratomas of comparable size (Fig. 4A,B), indicating no gross impact on tumorigenic potential. However, flow cytometry revealed a marked increase in polyploidy within teratomas derived from Chr2 + 3 and Chr2 + 1 cells (Fig. 4C), which was further confirmed by karyotype analysis (Fig. 4D,E).

To determine whether the in vivo–derived cells exhibited mitotic defects consistent with our in vitro observations, we re-

isolated tumor cells from teratomas and analyzed their mitotic behavior. Live-cell imaging revealed persistent chromosome lagging and frequent post-mitotic nuclear re-fusion events in cells derived from Chr2 + 3 and Chr2 + 1 lines (Fig. 4F,G). These findings indicate that translocation-induced mitotic instability and polyploidization are not limited to cultured pluripotent stem cells but also arise during in vivo differentiation across multiple somatic lineages.

## Discussion

Our study reveals a spindle-dependent biophysical constraint that imposes an upper limit on chromosome size in mammalian cells. When chromosome arms exceed the length of the spindle axis, segregation fails, leading to cytokinesis defects and progressive polyploidization. This size-dependent segregation failure is alleviated in tetraploid cells, where the mitotic spindle scales with genome content and restores proper chromosome segregation (Fig. 5).

These findings provide a mechanistic rationale for the evolutionary conservation of genome fragmentation in animals. While synthetic yeast can tolerate complete chromosomal fusion owing to its compact and structurally simple genome (Goffeau et al, 1996; Shao et al, 2018), mammalian genomes are substantially larger and distributed across multiple chromosomes. Even the smallest mouse chromosome (Chr19) contains more DNA than the entire *S. cerevisiae* genome (Liu et al, 2024; Waterston et al, 2002). Our data identify ~308 Mb (Chr4 + 5) as the maximal chromosome length compatible with faithful mitosis in diploid mouse ESCs, whereas longer constructs such as Chr2 + 3 (~340 Mb) exceed this threshold and undergo polyploidization. These results define a geometric boundary for mitotically permissible chromosome length in diploid cells.

This limit arises from a physical relationship between chromosome length and spindle geometry. Similar constraints have been documented in plant cells, where chromosome arms longer than half the spindle axis impair division (Hudakova et al, 2002; Schubert and Oud, 1997). Unlike plants constrained by rigid cell walls, mammalian cells divide through open mitosis and can dynamically adjust spindle length (Fowler and Quatrano, 1997; Goshima and Scholey, 2010). We show that this adaptability depends on Aurora B kinase, which extends the spindle to accommodate longer chromosomes. Inhibition of Aurora B collapses this adaptation, leading to segregation failure even in otherwise stable configurations such as Chr4 + 5. Thus, Aurora B functions as a mechanical coordinator that aligns spindle dimensions with chromosomal load to safeguard mitotic fidelity.

Interestingly, our earlier chromosome-fusion studies in mice provide in vivo support for this principle (Wang et al, 2022). Mice carrying the Chr4 + 5 fusion were viable and fertile, producing offspring with the same rearrangement, whereas embryos harboring the longer Chr1 + 2 fusion (~377 Mb) consistently arrested at E12.5. The only surviving line carried a spontaneous truncation that reduced Chr1 + 2 to ~263 Mb—below the ~308 Mb threshold identified here. At the time, the mechanistic basis of this developmental disparity was unclear; the current findings suggest that embryonic viability may depend on an intrinsic geometric limit to chromosome segregation. These data collectively indicate that spindle architecture, rather than external selection pressure,

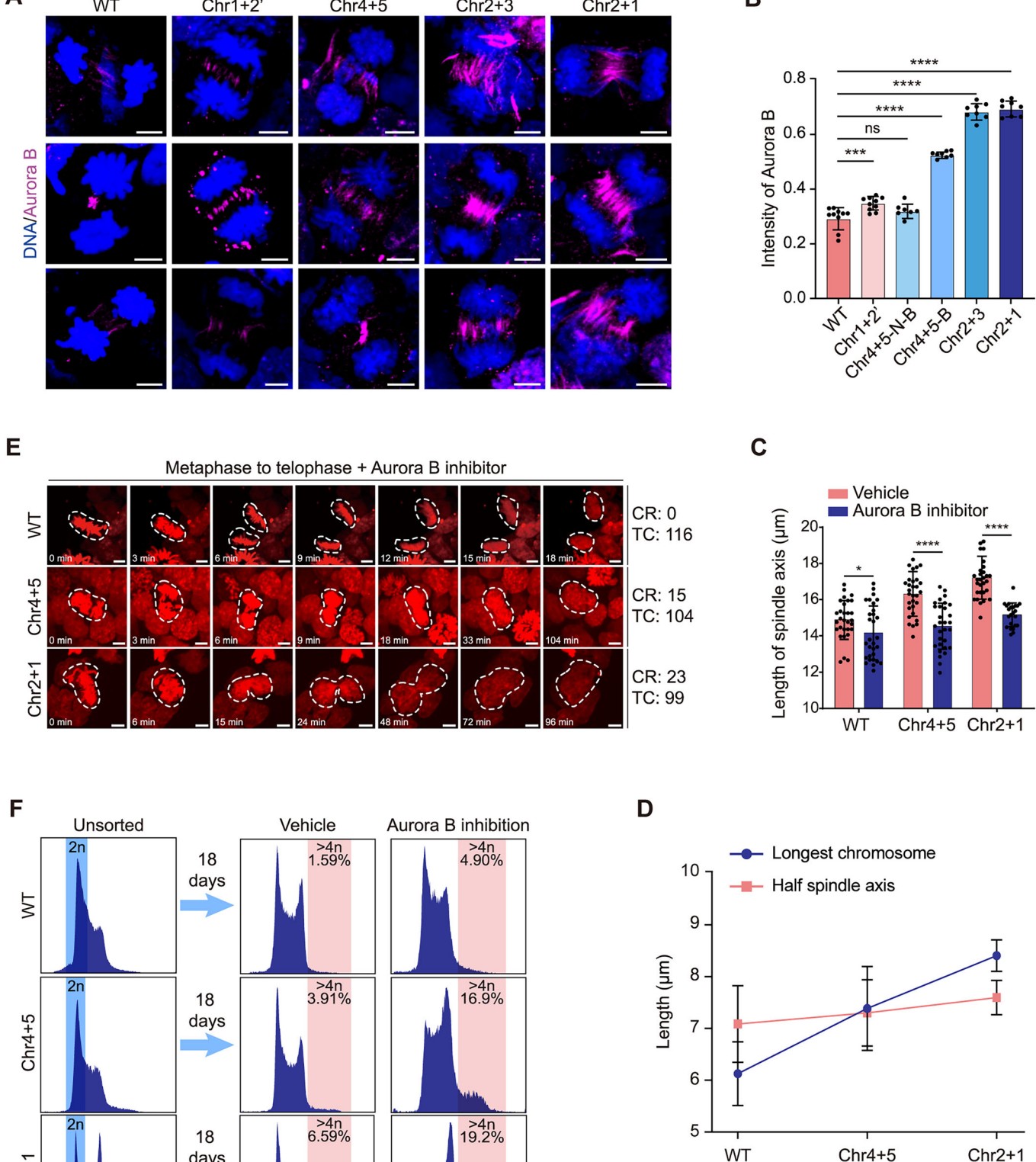

◀ **Figure 3.  Aurora B activity modulates spindle adaptation to chromosomal changes.**

(A) Aurora B localization (purple) and chromatin (blue) at the end of anaphase. Scale bar = 5 µm. (B) Quantification of Aurora B at the end of anaphase. WT: $n = 10$; Chr1 + 2': $n = 10$; Chr4 + 5-N-B (Chr4 + 5 without bridge): $n = 7$; Chr4 + 5-B (Chr4 + 5 with bridge): $n = 8$; Chr2 + 3: $n = 8$; Chr2 + 1: $n = 8$. Each data point shows the amount of Aurora B relative to that of chromatin. Statistical analysis was performed using unpaired $t$ test. Data are presented as mean ± SD, with a ns $P$ value of 0.1088, ***$P$ value of 2.06e-04, and ****$P$ value of 4.5e-05, 4.6e-05, 4.6e-05 rightward correspondingly. (C) Spindle axis measurements with/without Aurora B inhibitor. WT + Vehicle: $n = 30$; WT + Aurora B inhibitor: $n = 30$; Chr4 + 5 + Vehicle: $n = 30$; Chr4 + 5 + Aurora B inhibitor: $n = 30$; Chr2 + 1 + Vehicle: $n = 30$; Chr2 + 1 + Aurora B inhibitor: $n = 23$. Each data point represents the measurement length of the spindle axis. Statistical analysis was performed using unpaired $t$ test. Data are presented as mean ± SD, with *$P$ value of 0.0447, and ****$P$ value of 1.4e-06 and 3.0e-11 rightward, respectively. (D) Correlation between the longest chromosome length and half of the spindle axis length in WT, Chr4 + 5, and Chr2 + 1 diESCs with Aurora B inhibitor. Spindle axis length data are from (C), and chromosome length data are from Fig. EV4B. Data are presented as mean ± SD. (E) Mitotic progression in Aurora B-inhibited WT, Chr4 + 5, and Chr2 + 1 diESCs. White dotted circles indicate cells undergoing mitosis. For each group, numbers for cell re-coalescence (CR) and total observed cells (TC) represent pooled cell counts from multiple independent live-cell imaging experiments, including cells with complete mitotic progression. Scale bar = 5 µm. (F) Analysis of ploidy after Aurora B inhibition in WT, Chr4 + 5 and Chr2 + 1 diESCs. Blue rectangle: sorting of cells with DNA content equal to 2n. Pink rectangle: analyses of cells with DNA content > 4n. Source data are available online for this figure.

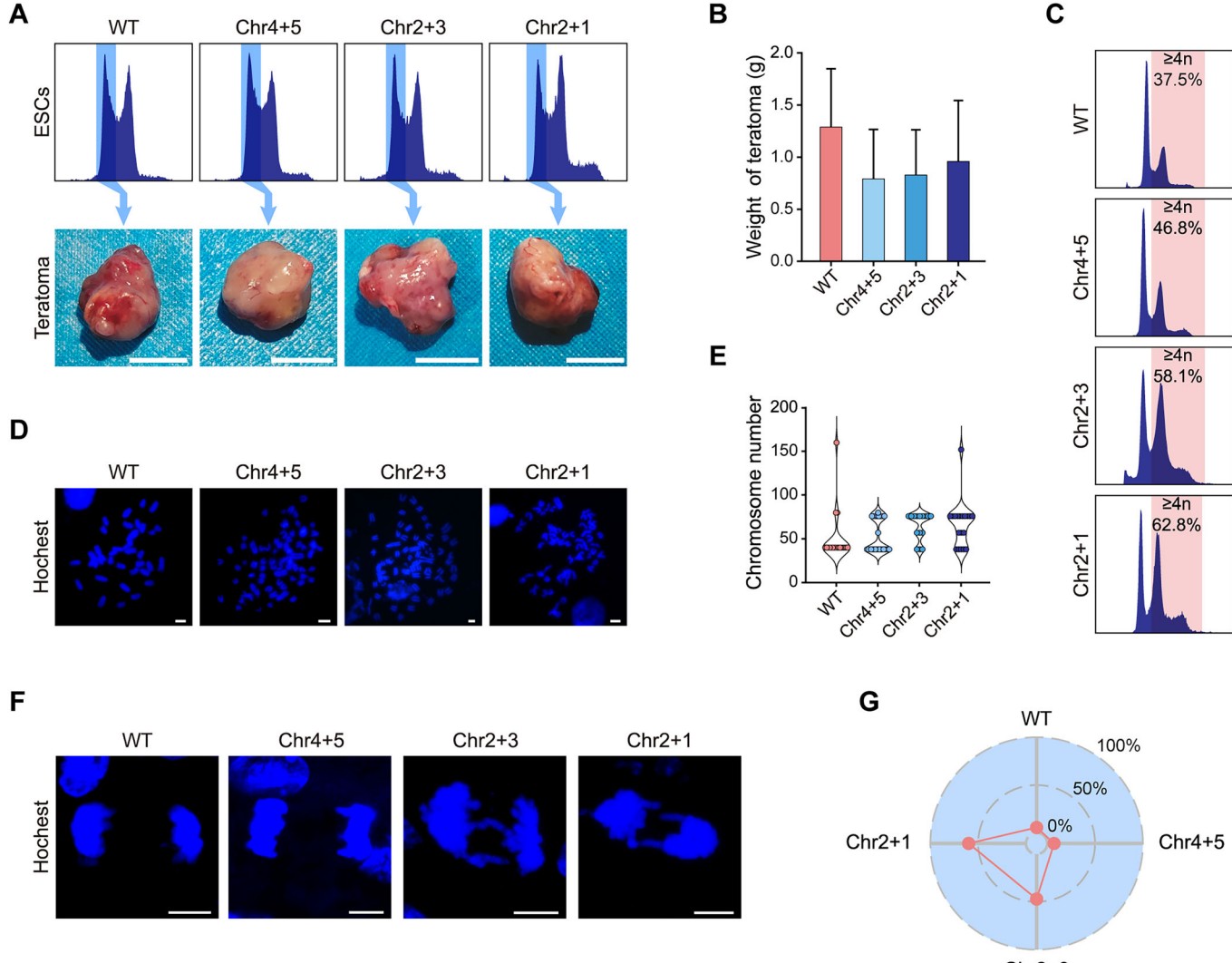

**Figure 4.  In vivo polyploidization in teratoma models.**

(A) Teratomas derived from WT and chromosome-translocated diESCs. Top: 2n cell sorting strategy. Bottom: teratomas from WT and chromosome-translocated diESCs. Scale bar = 5 mm. (B) Teratoma weights. Each bar represents the tumor weights of three biological replicates ($n = 3$). Data are presented as mean ± SD. (C) Ploidy distribution of teratoma-derived cells with DNA content ≥4n being indicated. (D) Representative karyotypes of teratoma cells. Scale bar = 5 µm. (E) Chromosome counts in teratoma cells (WT: $n = 19$; Chr4 + 5: $n = 35$; Chr2 + 3: $n = 15$; Chr2 + 1: $n = 19$). (F) Nuclear imaging of the WT, Chr4 + 5, Chr2 + 3, and Chr2 + 1 teratoma cells at the end of anaphase. Scale bar = 5 µm. (G) Radar chart of chromosome bridge frequency. Red dots represent the proportion of chromosome bridges (WT: 1/18; Chr4 + 5: 1/14; Chr2 + 3: 8/17; Chr2 + 1: 12/20). Source data are available online for this figure.

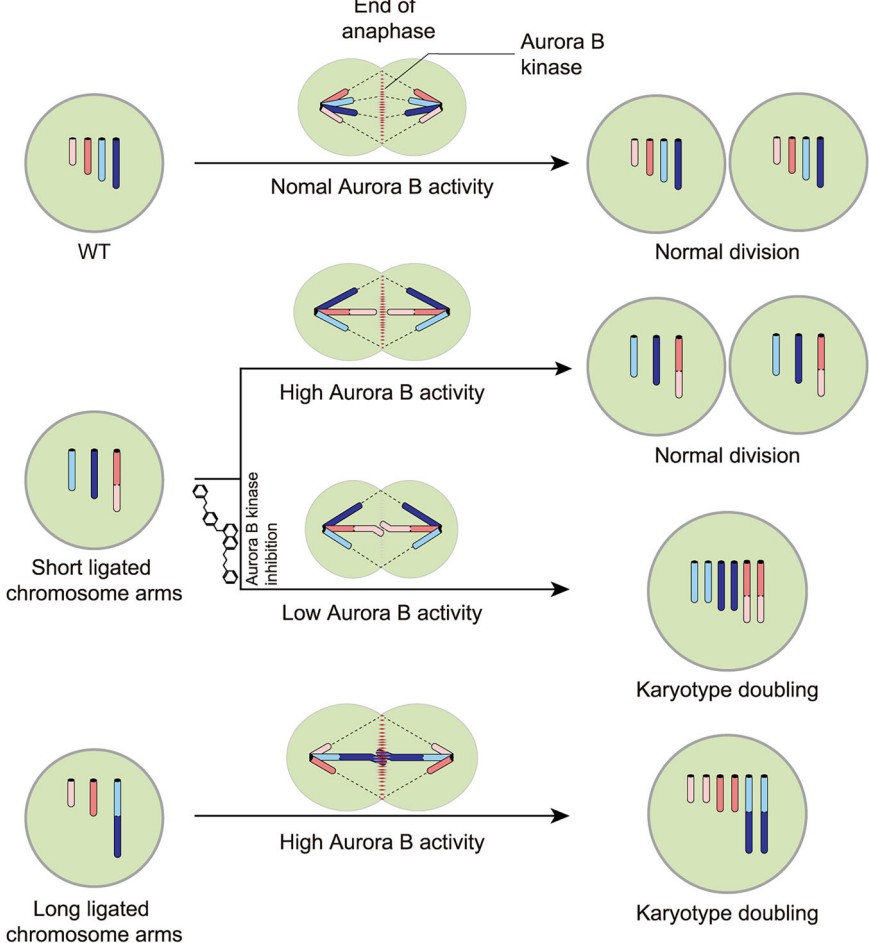

**Figure 5. Proposed model of mitotic fidelity regulated by chromosome–spindle scaling.**

Schematic illustrating how discordance between chromosome length and spindle axis dimensions leads to segregation errors and polyploidization in chromosome-translocated cells. The different color intensities reflect the activity of Aurora B, with darker colors indicating higher activity.

could represent a previously overlooked internal constraint shaping chromosome number and size across evolution.

Over evolutionary timescales, chromosomal fusions and translocations have occurred repeatedly in vertebrate lineages, yet no extant species possesses a single giant chromosome (Stapley et al, 2017). Instead, animals maintain a relatively narrow size distribution—typically spanning less than one order of magnitude—consistent with a conserved physical boundary (Li et al, 2011). Exceptional cases, such as lungfish whose ~18 chromosomes each exceed the size of the human genome, demonstrate that extreme genome expansion can be accommodated by proportional increases in cell and spindle size (Schartl et al, 2024). In our system, tetraploidization produced a similar adaptation: spindle length expanded with cell volume, allowing accurate segregation of longer chromosomes. Because genome doubling is a recurrent event in both plant and animal evolution, this interplay between cell size, spindle capacity, and chromosomal architecture may represent a hidden yet pervasive feedback mechanism driving karyotype evolution.

From this evolutionary perspective, the same geometric principles may also operate at shorter biological timescales. Polyploidization is not only an evolutionary process but also a common feature of normal development, tissue regeneration, and tumorigenesis (Shih et al, 2023; Soltis et al, 2014; Ullah et al, 2009). The biophysical constraints governing spindle–chromosome compatibility may therefore reappear in these contexts, influencing how cells tolerate genome duplication or structural rearrangements. In this light, Aurora B emerges as a potential vulnerability when this balance is perturbed, particularly in cells with abnormal karyotypes or ploidy states (Coschi et al, 2010; Gully et al, 2010; Wilkinson et al, 2007).

Beyond its mechanistic implications, our study also provides a conceptual and quantitative framework for synthetic genome engineering in mammals. In previous work, haploid ESCs were used as a versatile platform for generating engineered chromosome fusions (Wang et al, 2022). Here, by defining the upper physical limit of chromosome size tolerable by diploid mammalian cells, we extend this approach to the design space of mammalian karyotypes. The capacity of Chr4 + 5 to accommodate 308 Mb—~58% larger than the native Chr1—illustrates the inherent robustness of the mammalian mitotic machinery. This principle may guide future efforts to build synthetic chromosomes, enable chromosome-scale gene delivery, and rationally redesign mammalian genomes.

# Methods

### Reagents and tools table

| Reagent/resource | Reference or source | Identifier or catalog number |
|---|---|---|
| **Experimental models** | | |
| B6D2F1 (*M. musculus*) | Beijing Vital River | N/A |
| C57BL/6 (*M. musculus*) | Beijing Vital River | N/A |
| Haploid embryonic stem cells (*M. musculus*) | Li et al (2018) | N/A |
| Chr1 + 2′ embryonic stem cells (*M. musculus*) | Wang et al (2022) | N/A |
| Chr4 + 5 embryonic stem cells (*M. musculus*) | This study | N/A |
| Chr2 + 3 embryonic stem cells (*M. musculus*) | This study | N/A |
| Chr2 + 1 embryonic stem cells (*M. musculus*) | This study | N/A |
| SCID/Beige mice (*M. musculus*) | Charles River Laboratories | N/A |
| **Recombinant DNA** | | |
| pEASY-T1 | Transgen | Cat #CT101 |
| pUC19-U6-sgRNA | This study | N/A |
| pUC19-CAGp-SpCas9-2A-RFP-SV40polyA | Wang et al (2022) | N/A |
| PB-H2B-mCherry | Wang et al (2022) | N/A |
| **Antibodies** | | |
| Rabbit anti-Aurora B | Sigma | Cat #A5102 |
| Mouse anti-TUBULIN | Abcam | Cat #ab7291 |
| Rabbit anti-Lamin B1 | Abcam | Cat #ab133741 |
| Rabbit anti-β-Actin | Abclonal | Cat #AC026 |
| 594-Phalloidin | UElandy | Cat #YP0052S |
| **Oligonucleotides and other sequence-based reagents** | | |
| SgRNA for Chr2 + 3 fusion Chr2-tel-sgRNA: GTAGTATGCTGTCTGTTTGA**TGG**; Chr3-cen-sgRNA: GCCTCTGAAAACTATTGGAC**TGG** | This study | N/A |
| Primers for Chr2 + 3 fusion Forward: TACTGTGACCGAAGCA CATAACCAC; Reverse: GAGCAGGAAACTGTT CCTCTTCACA | This study | N/A |
| **Chemicals, enzymes, and other reagents** | | |
| CHIR99021 | Stemgent | Cat #04-0004 |
| PD0325901 | Stemgent | Cat #04-0006 |
| mLIF | Millipore | Cat #ESG1107 |
| Srcl2 | Sigma | Cat #255521 |
| PMSG | ProSpec | Cat #HOR-272 |
| hCG | ProSpec | Cat #HOR-250 |
| M2 | Sigma | Cat #M7167 |
| M16 | Sigma | Cat #M7292 |
| Paraffin oil | Sigma | Cat #M8410-1L |
| KOSM | Millipore | Cat #MR-020P-D |
| Hoechst 33342 | Invitrogen | Cat #H3570 |
| Cytochalasin B | Abeam | Cat #Ab143482 |
| knockout serum replacement | Gibco | Cat #10828028 |
| ZM447439 | MedChemExpress | Cat #HY-10128 |

| Reagent/resource | Reference or source | Identifier or catalog number |
|---|---|---|
| Colcemid | ThermoFisher | Cat #15212012 |
| KOD One™ PCR Master Mix -Blue | TOYOBO | Cat #KMM-201 |
| DMEM | Gibco | Cat #C11995500BT |
| DMEM/F12 | Gibco | Cat #11330-032 |
| Neurbasal | Gibco | Cat #21103-049 |
| FBS | Gibco | Cat #10099141 |
| Trypsin-EDTA(0.05%) | Gibco | Cat #25300062 |
| N2 | Gibco | Cat #17502048 |
| B27 | Gibco | Cat #17504044 |
| Penicillin-Streptomycin | Gibco | Cat #15140-122 |
| GlutaMAX™ Supplement | Gibco | Cat #35050079 |
| insulin | Roche | Cat #11376497001 |
| XMP 1 Green | Metasystems | Cat #D-1401-050-FI |
| XMP 2 Orange | Metasystems | Cat #D-1402-050-OR |
| XMP 3 Green | Metasystems | Cat #D-1403-050-FI |
| XMP 4 Green | Metasystems | Cat #D-1404-050-FI |
| XMP 5 Orange | Metasystems | Cat #D-1405-050-OR |
| Formamide | Sigma | Cat #F9037 |
| **Software** | | |
| Cas-Designer | http://www.rgenome.net/cas-designer/ | |
| Cas-OFFinder | http://www.rgenome.net/cas-offinder/ | |
| KTa-003 | GeneDiagnostics, Inc | |
| FlowJo | https://www.flowjo.com | V10.8.1 |
| Imaris | https://imaris.oxinst.com | V9.2 |
| **Other** | | |
| In-Fusion HD Cloning Kit | Clontech Laboratories | Cat #639649 |
| Mouse Direct PCR Kit | Bimake | Cat #B40015 |
| Cell Cycle and Apoptosis Analysis Kit | Yeasen Biotechnology | Cat #40301 |

PAM sequences are indicated in bold.

## Methods and protocols

### Animals care and use

All animal experiments were approved by the Institutional Animal Care and Use Committee of the Institute of Zoology, Chinese Academy of Sciences (IOZ, CAS), and the reference number for approval is IOZ20190074. Specific-pathogen-free (SPF) mice were obtained from Beijing Vital River Laboratories and subsequently raised in the animal facilities of IOZ, CAS. Chicken β-actin–GFP transgenic mice (B6D2F1× C57BL/6) were used to provide MII oocytes to derive haploid embryonic stem cells (haESCs) (Li et al, 2012). Male SCID/Beige mice (6 weeks old) were purchased from Charles River Laboratories (Wilmington, MA).

### Cell line establishment and culture

haESCs were derived from activated parthenogenetic embryos at E3.5 and cultured on feeder cells in 2i medium plus 5% knockout serum replacement (Gibco, 10828028) as described previously (Li et al, 2012). 2i medium consists of N2B27 medium supplemented with 1 µM MEK inhibitor PD0325901 (Stemgent, 04-0006), 3 µM GSK3b inhibitor CHIR99021 (Stemgent, 04-0004), and $10^3$ units/mL mLIF (Millipore, ESG1107). After 6–7 passages, the haESCs were purified by fluorescence-activated cell sorting (FACS). The haESCs were digested and incubated with 10 µg/mL Hoechst 33342 (ThermoFisher Scientific, H3570) at 37 °C for 15–20 min, followed by sorting and analysis using the Moflo Astrios EQ (Beckman). The 1n peak of DNA content was utilized for sorting haploid cells, while the 4n peak was employed for sorting diploid cells.

To inhibit Aurora B kinase activity, 2 µM ZM447439 (MedChemExpress, HY-10128) was added to 2i medium. To establish tetraploid ESCs from WT diESCs, cells with DNA content exceeding 4n were isolated using flow cytometry following a 24-hour treatment with 50 ng/mL Colcemid (ThermoFisher Scientific, 15212012).

All cell lines were authenticated and routinely tested negative for mycoplasma contamination.

### Plasmid construction

To induce programmed DNA cleavages in this study, the single-guide RNA (sgRNA)-guided *Streptococcus pyogenes* Cas9 (SpCas9) system was employed. All sgRNAs were designed using online software Cas-Designer (http://www.rgenome.net/cas-designer/) and Cas-OFFinder (http://www.rgenome.net/cas-offinder/). Only sgRNAs with evaluation scores above 66, no mismatches in the mouse genome, and a minimum distance from the repetitive sequences near centromeres and telomeres on the targeted chromosomes were selected. The unique targeting sites of the chosen sgRNAs were further verified using a BLAT search. Each sgRNA was cloned into a BsaI-digested pUC19-U6-sgRNA vector by ligating annealed oligonucleotides. For SpCas9 plasmid construction, the CAG promoter (CAGp), SpCas9, 2A-RFP and SV40polyA were synthesized and assembled by hierarchical fusion PCR. The resulting product, CAGp-SpCas9-2A-RFP-SV40polyA, was cloned into the pUC19 vector using In-Fusion HD Cloning Kit (Clontech Laboratories, 639649).

### CRISPR/Cas9-mediated chromosome ligation and PCR detection

To engineer the cells' chromosomes ligation, plasmids encoding sgRNAs and SpCas9 were co-transfected into haESCs using the Neon Transfection System (Invitrogen) following the manufacturer's instructions. Briefly, 110 µL of cell suspension ($2 \times 10^7$–$4 \times 10^7$ cells/mL) was incubated with 4 µg of SpCas9 plasmid and 4 µg of sgRNA plasmid, and transfected using 100 µL Neon tips (1300 V, 10 ms, and 3 pulses). The transfected haESCs were then plated onto feeder cells in 2i medium supplemented with 5% knockout serum replacement. Two days later, successfully transfected haESCs were sorted based on fluorescence expression, and subsequently plated at a low density (1000 cells/well) onto six-well plates pre-coated with feeder cells. After 7 or 8 days of culture, single clones were picked onto 96-well plates for expansion and PCR detection.

Chromosome ligation was detected by PCR with the KOD One™ PCR Master Mix -Blue (TOYOBO, KMM-201), and genomic DNA extracted from cells served as templates. Due to the highly repetitive sequences near telomeres or centromeres, genotyping primers with annealing temperatures higher than 60 °C were designed. Genomic DNA was extracted by the Mouse Direct PCR Kit (Bimake, B40015) following the manufacturer's instructions. Briefly, cells were resuspended by lysis buffer (Buffer L:Protease Plus = 50:1) and incubated at 55 °C for 2 h, followed 100 °C for 10 min. The PCR reaction procedure was as follows: 98 °C for 5 min; (94 °C for 15 s, Tm°C for 5 s, 68 °C for 1 kb/5 s) for 40 cycles; 68 °C for 10 min. The PCR products were detected with 2% agarose gel electrophoresis, and the results were validated by Sanger sequencing. The successfully identified chromosome-ligated ES cell lines were termed Chr2 + 1, Chr2 + 3, and Chr4 + 5.

### Karyotyping

ESCs and teratoma cells were treated with 50 ng/mL Colcemid for two to six hours. Cells were digested into single cell, collected by centrifugation, and washed once with PBS. The resulting pellets were resuspended in 200 µL DMEM, and then were hypotonically treated gently with 10 mL hypotonic solution (0.075 M KCl) at 37 °C for 30 min. The cells were then fixed with 1 mL fix solution (methanol: glacial acetic acid = 3:1) at room temperature (RT) for 5 min. After centrifugation, the pellets were resuspended and fixed with 6 mL cold fresh fix solution for another 20 min. This step was repeated once. Then the pellets were collected by centrifugation and resuspended in 200 µL cold fix solution. Cell solution (10 µL per drop) was dropped upon cold glass slides (pre-cooled at −20 °C or −80 °C) by gravity, and chromosomes were spread. For karyotype observation, the slides were stained with Hoechst 33342 and observed using the LSM780 Meta confocal microscope (Zeiss). All standard G-banding karyotyping results were analyzed with software KTa-003 (GeneDiagnostics, Inc.).

### Ploidy analyses of cells conceiving ligated chromosomes

To investigate potential polyploidization, the chromosome-ligated ES cell lines were incubated with Hoechst 33342 at 37 °C for 15–20 min, and GFP-positive cells with 2n DNA content were sorted by FACS. The sorted 2n ESCs were seeded onto six-well plates at a density of $10^4$ cells/well. After 26 days of culture, the DNA content of ≥4n ESCs was analyzed by FACS. After 18 days of Aurora treatment of WT, Chr4 + 5 and Chr2 + 1 cell lines, the DNA content of å 4n ESCs was analyzed by FACS.

To investigate whether chromosome ploidy continues to increase after cell quadruploidy, GFP-positive cells with 8n DNA content were sorted by FACS and seeded onto six-well plates at a density of $10^4$ cells/well. After 26 days of culture, the DNA content of >8n ESCs was analyzed by FACS. Data were collected by Beckman MoFlow XDP II and analyzed using the FlowJo software.

### Cell cycle analyses

To conduct the cell cycle, the Cell Cycle and Apoptosis Analysis Kit (Yeasen Biotechnology, 40301) was used. In brief, the ESCs were digested and fixed with 70% ethanol at 4 °C overnight followed by washing with PBS. According to the manufacturer's instructions, add 10 µL propidium iodide storage solution (40301-B) and 10 µL RNaseA (40301-A) solution into 0.5 mL dyeing buffer (40301-C), mixed well, added to the cell sample, incubated at 37 °C for 30 min. FACS was performed on the Beckman MoFlow XDP II, and the data were analyzed by FlowJo software (FlowJo LLC).

### Live-cell imaging and measurements of chromosome length and movement distances

Live-cell imaging was conducted using the Andor Dragonfly 200 system. All types of chromosome-ligated cells and wild-type (WT) cells were labeled with the PB-H2B-mCherry plasmid, as previously described (Wang et al, 2022). For analysis of chromosome separation, images were taken every 3 min with z stacks for 4 to 5 h. Mitotic cells were identified based on chromosome morphology. The longest chromosome length and the chromosome movement distance at the end of anaphase were measured using the software Imaris 9.2 image analysis. No blinding was performed.

### Immunofluorescence staining and fluorescence intensity measurement

Immunofluorescence staining was carried out as previously described (Zhao et al, 2009). In brief, samples were fixed with 4% paraformaldehyde at RT for 30 min and then washed twice with PBS. To block non-specific binding sites, 400 μL of 2% BSA plus 0.3% Triton X-100 was applied at RT for 1 h. The samples were then incubated with primary antibodies at 4 °C overnight. Following incubation, samples were washed three times with PBS, incubated in a Cy3-AffiniPure secondary antibody at RT for 1 h, and washed three times with PBS. Then the nuclei were stained by 10 μg/mL Hoechst 33342 at RT for 10 min, and imaged using the TCS Sp8 confocal microscope (Leica).

Fluorescence intensities of Aurora B and DAPI were quantified using Imaris software (version 9.2). The relative fluorescence intensity of Aurora B was normalized to that of DAPI for each image. No blinding was performed.

### Teratoma experiments

After sedimentation for 30 min at 37 °C to remove the feeder, 1E7 ESCs in 100 μL PBS suspension were subcutaneously injected into one side of the dorsal flanks of SCID/Beige mice. Approximately 3–4 weeks after injection, when the tumor diameter reached 1.5 cm, primary teratomas were removed. Mice that survived allowed the prolonged observation of metastasis.

For immunohistochemical (IHC) staining, primary teratomas were partially sectioned and fixed in 10% buffered formalin phosphate for 24 h, after which they were embedded in paraffin for tissue sectioning.

### Primary teratoma cell preparation and analysis

Teratoma tissues were washed with ice-cold PBS, cut, and digested at 37 °C for 5 min. The digestion was stopped by adding DMEM with 10% FBS, followed by centrifugation ($300\times g$, 5 min). Tissue fragments were cultured in six-well plates with DMEM containing 10% FBS and antibiotics (penicillin/streptomycin) at 37 °C under 5% $CO_2$ for 4–5 days. Then the Primary Teratoma Cells were digested and incubated with 10 μg/mL Hoechst 33342 (Thermo-Fisher Scientific, H3570) at 37 °C for 15–20 min, followed by sorting and analysis using the Moflo Astrios EQ (Beckman).

### DNA FISH

DNA fish was carried out as previously described (Wang et al, 2022). In brief, ES cells were fixed with 4% paraformaldehyde at RT for 15 min and then permeated with membrane permeation solution 1 (0.2% Triton X-100 (Sigma, T9284)) at RT for 30 min and then membrane permeation solution 2 (0.7% Triton X-100 and 0.1 M Hydrochloric acid) on ice for 15 min. After permeation, samples were washed twice with 70% alcohol, and dehydration was performed in 80%, 95%, 100%, 100% alcohol at RT for 3 min, respectively. Slices were then air-dried for 5 min. After drying, add the hybridization solution (50% Formamide (Sigma, F9037) and 20× SSC) and incubate at 80 °C for 30 min, and wash twice with pre-cooled anhydrous ethanol. Then, the probe mixture (XMP 1 Green and XMP 2 Orang; XMP 2 Orange and XMP 3 Green; or XMP 4 Green and XMP 5 orange probe mixture, all probes from Metasystems) was denatured at 95 °C for 2 min and then quickly placed on ice. For chromosome painting, cover the samples with the probe mixture and cover with a coverslip. Samples were then hybridized in a humidified incubator at 37 °C overnight. After hybridization, all slices were performed 5-min-washing for three times by hybridization washing solution (50% formamide, 2× SSC) at 42 °C and 5-min-washing for four times with 2× SSC. Each slice was stained with 10 μg/mL Hoechst 33342 for 10 min at RT and then was imaged with the TCS Sp8 confocal microscope (Leica).

### RNA-seq data analysis

Using one newly sequenced RNA-seq data from Chr4 + 5 diESCs ("Data availability") and our previously obtained RNA-seq data (Data ref: Wang et al, 2022), we controlled and trimmed the reads of WT, Chr1 + 2', Chr4 + 5, and Chr2 + 1 ESCs with fastqc (V0.11.5) and cutadapt (V0.39). Using Hisat2 (version 2.1.0), sequencing data were mapped to the mm10 reference genome (Kim et al, 2019). To compare the gene expression profile of our RNA-seq data, we normalized our data using Fragments Per Kilobase of transcript per Million (FPKM) normalization. The differential gene expression profile, which included genes with "fold change ≥ 2, $q \leq 0.05$", was computed using DESeq2 (Love et al, 2014).

### Quantification and statistical analysis

Statistical details consisting of $N$, mean, and statistical significance values are listed in the text or figure legends. Error bars in the figures represent standard deviation (SD) from independent experiments or independent samples. In all figures, one, two, three, and four asterisks represent $*P < 0.05$, $**P < 0.01$, $***P < 0.001$, and $****P < 0.0001$, respectively; ns represents not significant.

# Data availability

The datasets produced in this study are available in the following database: RNA-seq data: Genome Sequence Archive CRA030371.

The source data of this paper are collected in the following database record: biostudies:S-SCDT-10_1038-S44320-026-00188-8.

# Peer review information

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

## Acknowledgements

We thank Q Meng, X Yang, X-L Zhu, and S-W Li from the Institute of Zoology, Chinese Academy of Sciences, for their help with fluorescence-activated cell sorting and confocal laser-scanning microscopy. This work is supported by the National Key Research and Development Program (2024YFA0917403 to L-BW and 2022YFA1103600 to LW), Beijing Natural Science Foundation (7242083 to L-BW and Z230011 to WL), the National Natural Science Foundation of China (32470910 to L-BW and 32225030 to WL), Open Research Fund of National Health Commission Key Laboratory of Birth Defects Prevention (NHCKLBDP202409).

## Author contributions

**Yu-Long Zhao**: Data curation; Writing—original draft. **Yi-Ming Zhao**: Data curation. **Yi-Fang Yan**: Data curation. **Ning Yang**: Data curation. **Si-Nan Ma**: Data curation. **Rui-Jia Wang**: Data curation. **Gui-Hai Feng**: Data curation. **Zhi-Kun Li**: Conceptualization; Investigation; Writing—review and editing. **Wei Li**: Funding acquisition; Writing—review and editing. **Li-Bin Wang**: Conceptualization; Data curation; Supervision; Funding acquisition;

Investigation; Writing—original draft; Project administration; Writing—review and editing.

Source data underlying figure panels in this paper may have individual authorship assigned. Where available, figure panel/source data authorship is listed in the following database record: biostudies:S-SCDT-10_1038-S44320-026-00188-8.

## Disclosure and competing interests statement

The authors declare no competing interests.

# Expanded View Figures

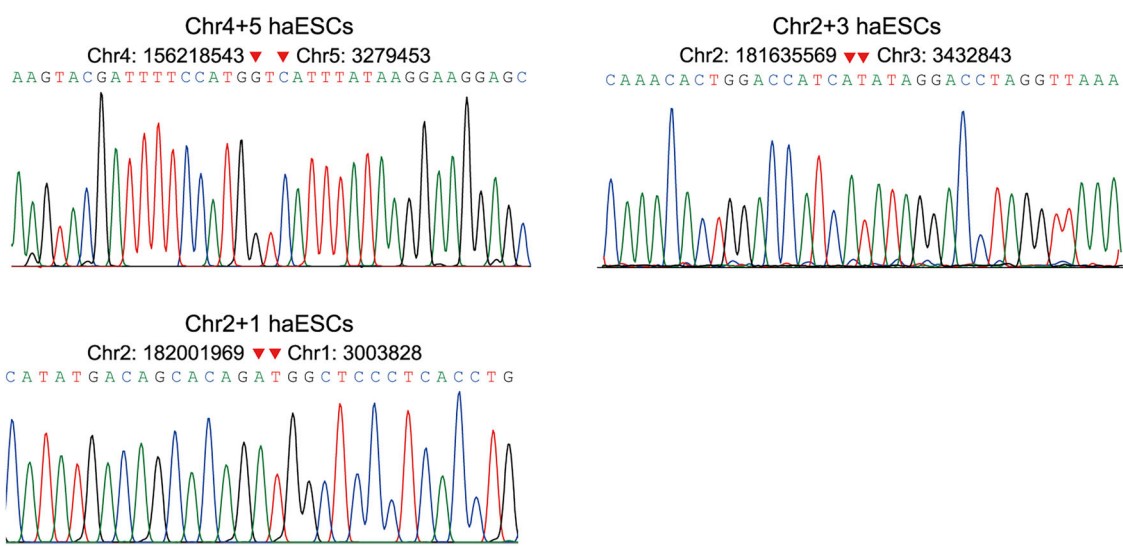

**Figure EV1.  Validation of chromosomal translocations in haploid embryonic stem cells.**

Sanger sequencing verification of translocation breakpoint junctions. Red triangles denote fusion-site nucleotides at derivative chromosome termini.

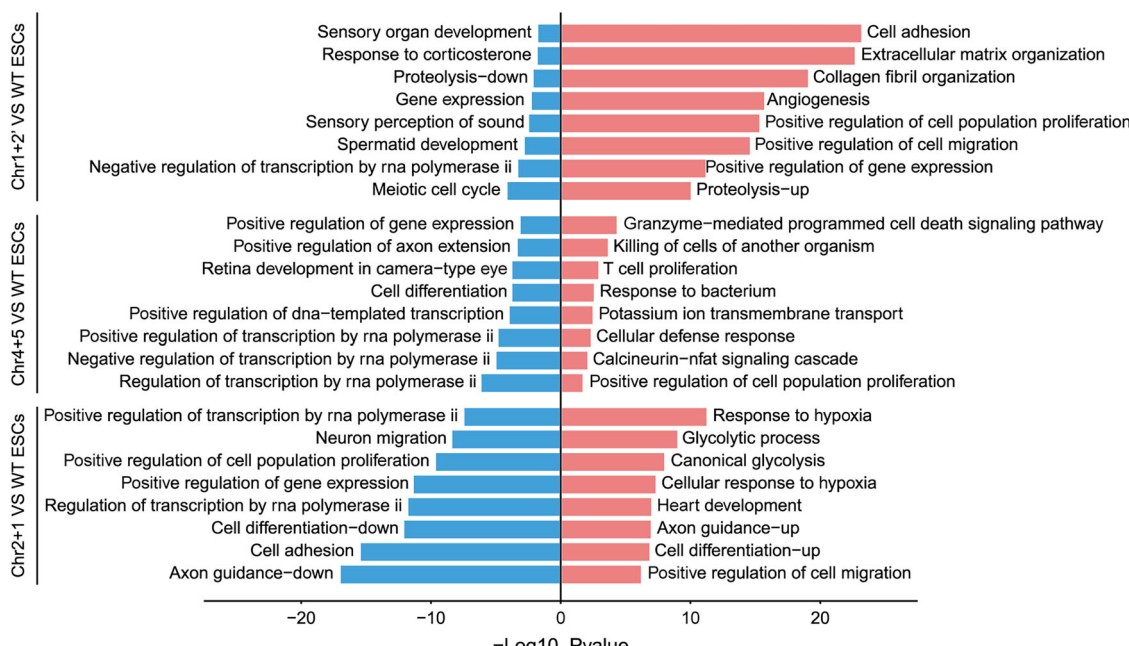

**Figure EV2.   Gene Ontology enrichment analysis of differentially expressed genes.**

Gene Ontology enrichment analysis of differentially expressed genes in chromosome-translocated diESCs (Chr1 + 2′, Chr4 + 5, and Chr2 + 1) compared to WT diESCs. The x axis indicates the $\log_{10}$-transformed FDR values, and the y axis lists the enriched pathway terms. Red bars represent upregulated pathways, and blue bars represent downregulated pathways. Enrichment analysis was calculated by using Fisher's exact test.

    

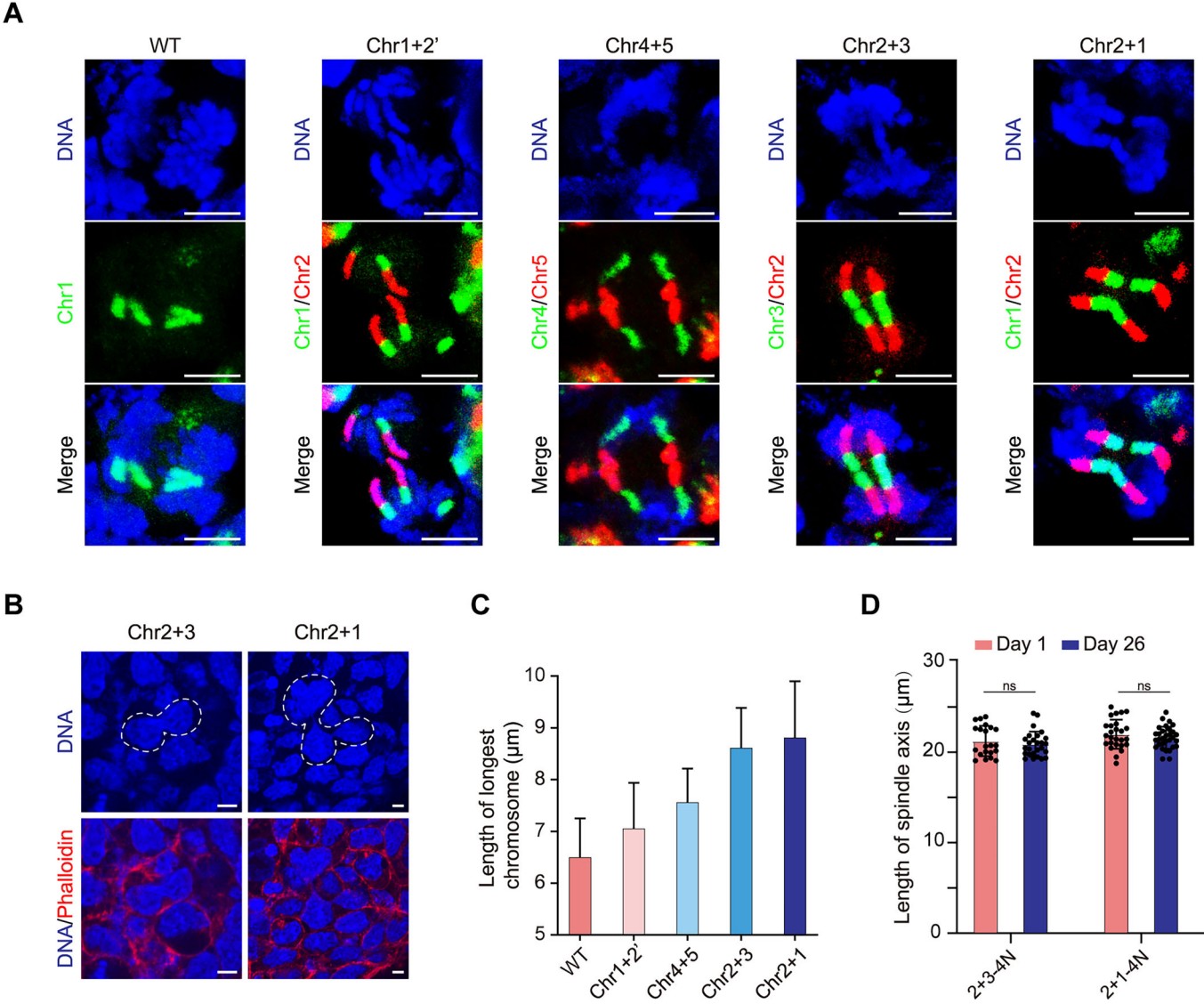

**Figure EV3. Observations and measurements of translocated chromosomes and spindles at anaphase.**

(A) FISH detection of lagging chromosomes at anaphase. Scale bar = 5 μm. (B) Immunofluorescence staining of Chr2 + 1 and Chr2 + 3 diESCs. Phalloidin staining indicates the spread of cytoskeleton, and white dotted circles label cells with binucleate and trinucleate. Scale bar = 5 μm. (C) Measurements of the longest chromosomes observed in live-cell imaging at the end of anaphase. WT: $n = 108$; Chr1 + 2': $n = 44$; Chr4 + 5: $n = 45$; Chr2 + 3: $n = 58$; Chr2 + 1: $n = 76$. Data are presented as mean ± SD. (D) Comparison of spindle axis in tetraploid cells on day 1 and day 26. Day 1 Chr2 + 3-4 N: $n = 21$; Day 26 Chr2 + 3-4 N: $n = 26$; Day1 Chr2 + 1-4 N: $n = 26$; Day 26 Chr2 + 1-4 N: $n = 31$. Each data point represents one spindle axis. Statistical analysis was performed using unpaired $t$ test. Data are presented as mean ± SD, with ns $P$ value of 0.6598 and 0.3701 rightward correspondingly.

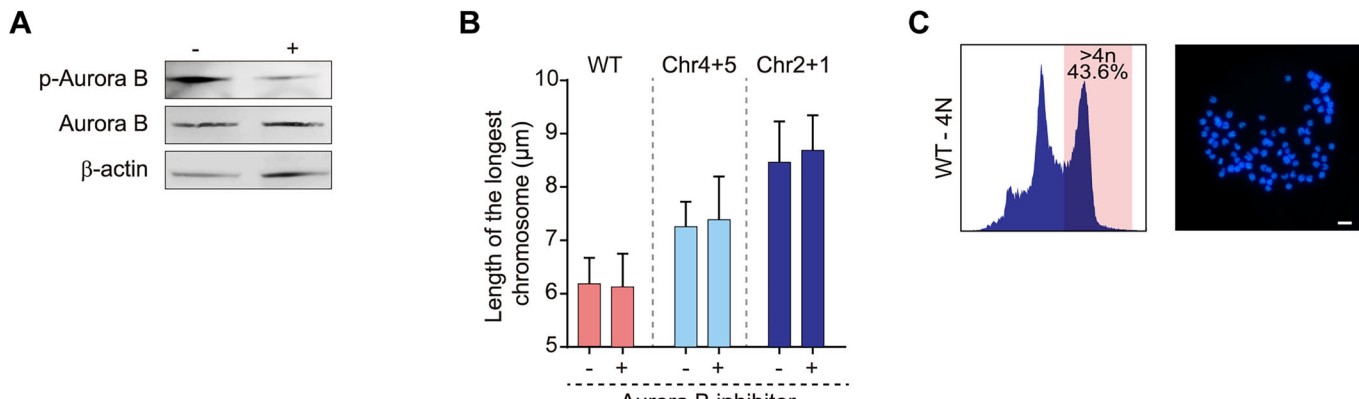

**Figure EV4. Chromosome length and ploidy in WT diESCs following Aurora B inhibition.**

(A) Immunoblot analysis of Aurora B autophosphorylation in WT diESCs with and without Aurora B inhibitor. β-actin serves as loading control. (B) Length comparison of the longest chromosomes in WT, Chr4 + 5 and Chr2 + 1 diESCs with and without Aurora B inhibitor. WT + Vehicle: $n = 59$; WT + Aurora B inhibitor: $n = 58$; Chr4 + 5 + Vehicle: $n = 57$; Chr4 + 5 + Aurora B inhibitor: $n = 59$; Chr2 + 1 + Vehicle: $n = 59$; Chr2 + 1 + Aurora B inhibitor: $n = 27$. Data are presented as mean ± SD. (C) FACS and karyotyping analysis of tetraploid WT ESCs. Tetraploid cells were isolated from WT diESCs following 18 days of Aurora B inhibition. The pink rectangle highlights the population with a DNA content >4n. Scale bar = 5 μm.

