## [Peer Review File · Molecular Systems Biology]

Chromosome length is constrained by spindle scaling to ensure faithful mitosis in mammals

Libin Wang, Yulong Zhao, Yiming Zhao, Ning Yang, Sinan Ma, Ruijia Wang, Guihai Feng, Zhi-kun Li, Wei Li, and Yifang Yan

Corresponding author(s): Libin Wang (wanglibin@bjmu.edu.cn), Wei Li (liwei@ioz.ac.cn), Guihai Feng (fenggh@ioz.ac.cn), Zhi-kun Li (lizhikun@ioz.ac.cn)

Review Timeline:

Submission Date:	12th Jun 25
Editorial Decision:	22nd Jul 25
Revision Received:	1st Nov 25
Editorial Decision:	11th Dec 25
Revision Received:	16th Dec 25
Accepted:	2nd Jan 26

Editor: Poonam Bheda

Transaction Report:

22nd Jul 2025

Manuscript Number: MSB-2025-13173

Title: Chromosome length is constrained by spindle scaling to ensure faithful mitosis in mammals

Dear Dr Wang,

Thank you for the submission of your manuscript to Molecular Systems Biology. We have now received feedback from the two/three reviewers who agreed to evaluate your manuscript. As you will see from the reports below, the referees acknowledge the interest of the study and are overall supportive of your work; however they also comment on multiple aspects of the manuscript that should be strengthened in a revision.

I think that the recommendations of the reviewers are rather clear and I therefore do not see the need to repeat the comments listed below. All of the issues raised would need to be satisfactorily addressed. Please let me know in case you would like to discuss in further detail any of the any of the reviewer comments or your proposed revisions, I would be happy to schedule a call.

We require:

1) A .docx formatted version of the manuscript text (including legends for main figures, EV figures and tables). Please make sure that the changes are highlighted to be clearly visible. Alternatively you may choose to submit your manuscript as a LaTeX file.

4) A .docx formatted letter INCLUDING the reviewers' reports and your detailed point-by-point responses to their comments. As part of the EMBO Press transparent editorial process, the point-by-point response is part of the Peer Review File (PRF), which will be published alongside your paper.

5) A complete author checklist, which you can download from our author guidelines (<https://www.embopress.org/page/journal/17574684/authorguide#submissionofrevisions>). Please insert information in the checklist that is also reflected in the manuscript. The completed author checklist will also be part of the PRF.

6) Please note that all corresponding authors are required to supply an ORCID ID for their name upon submission of a revised manuscript.

7) It is mandatory to include a 'Data Availability' section after the Materials and Methods. Before submitting your revision, primary datasets produced in this study need to be deposited in an appropriate public database, and the accession numbers and database listed under 'Data Availability'. Please remember to provide a reviewer password if the datasets are not yet public (see <https://www.embopress.org/page/journal/17574684/authorguide#dataavailability>).

In case you have no data that requires deposition in a public database, please state so in this section as follows: "This study includes no data deposited in external repositories". Note that the Data Availability Section is restricted to new primary data that are part of this study.

8) All Materials and Methods need to be described in the main text using our 'Structured Methods' format, which is required for all research articles. According to this format, the Methods section includes a Reagents and Tools Table (listing key reagents, experimental models, software and relevant equipment and including their sources and relevant identifiers) followed by a Methods and Protocols section describing the methods using a step-by-step protocol format. The aim is to facilitate adoption of the methodologies across labs. Please upload the Reagents and Tools table as a separate document when submitting your revised manuscript. More information on how to adhere to this format as well as a downloadable template (.docx) for the Reagents and Tools Table can be found in our author guidelines:

<https://www.embopress.org/page/journal/17444292/authorguide#structuredmethods>

An example of a Method paper with Structured Methods can be found here:
<https://www.embopress.org/doi/10.15252/msb.20178071>.

9) For data quantification: please specify the name of the statistical test used to generate error bars and p-values, the number (n) of independent experiments (specify technical or biological replicates) underlying each data point and the test used to calculate p-values in each figure legend. The figure legends should contain a basic description of n, p-values and the test applied. Graphs must include a description of the bars and the error bars (s.d., s.e.m.). Please provide exact p-values (in either the figure or figure legend).

10) Our journal encourages inclusion of *data citations in the reference list* to directly cite datasets that were re-used and obtained from public databases. Data citations in the article text are distinct from normal bibliographical citations and should directly link to the database records from which the data can be accessed. In the main text, data citations are formatted as follows: "Data ref: Smith et al, 2001" or "Data ref: NCBI Sequence Read Archive PRJNA342805, 2017". In the Reference list, data citations must be labeled with "[DATASET]". A data reference must provide the database name, accession number/identifiers and a resolvable link to the landing page from which the data can be accessed at the end of the reference. Further instructions are available at .

11) We replaced Supplementary Information with Expanded View (EV) Figures and Tables that are collapsible/expandable online. EV Figures should be cited as 'Figure EV1, Figure EV2' etc... in the text and their respective legends should be included in the main text after the legends of regular figures.

- Additional Tables/Datasets should be labeled and referred to as Table EV1, Dataset EV1, etc. Legends should be provided in a separate tab in case of .xls files. Alternatively, the legend can be supplied as a separate text file (README) and zipped together with the Table/Dataset file.

<https://www.embopress.org/page/journal/17574684/authorguide#expandedview>

12) Author contributions: CRediT has replaced the traditional author contributions section because it offers a systematic machine-readable author contributions format that allows for more effective research assessment. Please remove the Authors Contributions from the manuscript and use the free text boxes beneath each contributing author's name in our system to add specific details on the author's contribution. More information is available in our guide to authors.

13) Disclosure statement and competing interests: We updated our journal's competing interests policy in January 2022 and request authors to consider both actual and perceived competing interests. Please review the policy <https://www.embopress.org/competing-interests> and update your competing interests if necessary.

14) Every published paper now includes a 'Synopsis' to further enhance discoverability. Synopses are displayed on the journal webpage and are freely accessible to all readers. They include a short stand first (maximum of 300 characters, including space) as well as 2-5 one-sentences bullet points that summarizes the paper. Please write the bullet points to summarize the key NEW findings. They should be designed to be complementary to the abstract - i.e. not repeat the same text. We encourage inclusion of key acronyms and quantitative information (maximum of 30 words / bullet point). Please use the passive voice. Please attach these in a separate file or send them by email, we will incorporate them accordingly.

Please note that these would be the final versions and changes during proofing are usually not allowed.

15) As part of the EMBO Publications transparent editorial process initiative (see our policy here: https://www.embopress.org/transparent-process#Review_Process), Molecular Systems Biology will publish online a Peer Review File (PRF) to accompany accepted manuscripts.

In the event of acceptance, this file will be published in conjunction with your paper and will include the anonymous referee reports, your point-by-point response and all pertinent correspondence relating to the manuscript. Let us know whether you agree with the publication of the PRF and as here, if you want to remove or not any figures from it prior to publication. Please note that the Author checklist will be published at the end of the PRF.

Molecular Systems Biology has a "scooping protection" policy, whereby similar findings that are published by others during review or revision are not a criterion for rejection. Should you decide to submit a revised version, I do ask that you get in touch after three months if you have not completed it, to update us on the status.

Yours sincerely,

Reviewer #1:

Evaluation of the manuscript Chromosome length is constrained by spindle scaling to ensure faithful mitosis in mammals, MSB-2025-13173

The manuscript presents a very nice model system in which mouse stem cells were modified to contain one copy of an extra-large fusion chromosome. Characterization of mitosis shows that cells with the extra long chromosome often undergo whole-genome doubling, through somewhat unclear mechanisms. The authors show also that the spindle axis increases with the chromosome length, and this they attribute to Aurora B. Inhibition of Aurora B reduces the changes in length of the spindle axes in cells with fusion chromosomes, and causes an increased accumulation of cells with aberrant ploidy. Similar results can be observed also *in vivo*. This is a creative, original story, bringing up a very interesting hypothesis explaining the need to keep chromosome length below a certain threshold in order to enable efficient chromosome segregation during mitosis. However, there are several critical shortcomings, which need to be addressed should the manuscript be published.

1. In Fig. 1C, the authors claim that the cells showed elevated G2/M ratio. This could be interpreted that the ratio of G2 to M cells has changed, which is not what the authors show. The authors should rephrase the sentence.
2. The results of the transcriptomics are not very informative. Why was this done? Also, the sentence "This paradox implies that chromosomal rearrangements primarily induce mitotic stress through physical constraints rather than transcriptional reprogramming" is very confusing. What transcriptional reprogramming have the authors in mind which would lead to whole genome doubling?
3. The authors claim that Fig. 2a "...demonstrate complete failure of chromosomal segregation at anaphase termination". This is not shown in the figure at all.
4. In figure 2A, the authors claim that the bridges are due to the large chromosome, but they never show any evidence that the arising chromatin bridges are indeed the fusion chromosomes. This evidence has to be provided. Also, what fraction of cells shows chromatin anaphase bridges? Do all cells with bridges become tetraploid?
5. The authors claim that the chromatin bridges arising from fusion chromosomes lead to re-fusion of the chromosome masses into one tetraploid nucleus. This is highly unusual. Most of the time, chromatin bridges result in binucleated cells (e.g., Davoli, de Lange, 2012, Afonso et al, 2014). The authors cite the Afonso paper, and talk about cytokinesis failure, but this is not what they present in their figures.
6. The title in Fig. 2B Ratio of cell fusion is completely misleading, because there was certainly no cell fusion (meaning fusion between two independent cells) occurring.
7. How was the spindle length measured? There is no information in the methods, and none of the images shown in the figures would allow measuring spindle length.
8. Why do the tetraploid cells tolerate the long chromosomes? Do they have longer spindles? This has not been shown in the manuscript, and in fact, there are several papers suggesting that the spindle length does not scale with increasing ploidy (Storchova et al, 2006; Cohen-Sharir et al, 2021; Gudlin et al, 2025 and more). Did the length of spindle change from day 1 to day 26 in tetraploid ESCs (Fig. 1E)?
9. Fig. 3A shows example for Aurora B localization, but this is just one cell per sample, and no quantification can be used. This needs to be added. Also, the amount of Aurora B in spindles with and without bridges should be compared.
10. Aurora B inhibition is causing a lot of phenotypes. In fact, polyploidy upon treatment with Aurora B inhibitor is among the most usual phenotypes. Why is this not happening here in the wild type (Fig. 3E)? Why are the cell cycle profiles so different for Chr4+5 and Chr2+1 in Aurora B inhibition experiment (Fig. 3E)?
11. The figure 3D, Chr4+5 is basically indiscernible, it is not clear what it shows.
12. What is "increased chromosome load"? (last paragraphs of chapter Aurora B kinase regulates spindle adaptation).
13. The discussion is highly speculative. The authors had a nice idea and show some interesting data, but the data is not complete and allow other, potentially more interesting interpretations.

Reviewer #2:

The authors show here that chromosome length in mammalian cells is constrained by a biophysical limit governed by spindle geometry. Their findings address the structural basis for genome fragmentation in animals and provide a general mechanism linking chromosome size, spindle dynamics, and genome stability. The topic is of great importance and timely. However, there are a few concerns that should be addressed before its acceptance for publication.

1. The authors should annotate the phenotype of real-time imaging to visualize the chromosome damage using reports;
2. The authors argue for the length alternation and should perform unbiased quantification of telomere-telomere distance.

Reviewer #3:

I read the concise manuscript of Zhao and colleagues with great interest. Pioneering work of the group of Ingo Schubert has already shown more than 25 years ago that there seems to be an upper limit of chromosome size in plant cells. They found indications that the upper limit for a chromosome arm is half of spindle axis length. If arms are bigger mitotic failure occurs. The authors of the current manuscript pioneered recently chromosome engineering in mice by inducing chromosome fusions using CRISPR/Cas. As mouse chromosomes are telocentric by cuts in the centromere-distal telomer and next to the centromere of another chromosome fusion can be obtained. Thus, depending on length of the fusion partners different sized chromosome were produced. The authors obtained these set of fusions in ESCs. They found an arm length of about 308 Mb as maximum tolerated length for diploid cells. Interestingly in tetraploid cells longer arm sized were tolerated due to the fact that spindle axis is also larger in these cells. These findings nicely sustain Ingo Schubert's hypothesis and demonstrate that show that the limitation is not restricted to cells with a rigid cell wall. This is a very nice and well-done study and I do not have any suggestions to improve it.

Point-by-point response

We sincerely thank all reviewers for their careful evaluation and constructive suggestions. In response to the comments, we have conducted a series of additional experiments and analyses that substantially strengthen the evidence supporting our conclusions. Alongside these new data, we have refined the manuscript for improved precision, completeness, and accuracy, including clearer figure presentations, expanded methodological details, and a more coherent organization of the main text. These revisions have markedly enhanced the rigor, clarity, and interpretative depth of the study, and we are deeply grateful to the reviewers for their insightful feedback that guided these improvements.

Reviewer #1:

Evaluation of the manuscript Chromosome length is constrained by spindle scaling to ensure faithful mitosis in mammals, MSB-2025-13173

The manuscript presents a very nice model system in which mouse stem cells were modified to contain one copy of an extra-large fusion chromosome. Characterization of mitosis shows that cells with the extra long chromosome often undergo whole-genome doubling, through somewhat unclear mechanisms. The authors show also that the spindle axis increases with the chromosome length, and this they attribute to Aurora B. Inhibition of Aurora B reduces the changes in length of the spindle axes in cells with fusion chromosomes, and causes an increased accumulation of cells with aberrant ploidy. Similar results can be observed also in vivo. This is a creative,

original story, bringing up a very interesting hypothesis explaining the need to keep chromosome length below a certain threshold in order to enable efficient chromosome segregation during mitosis. However, there are several critical shortcomings, which need to be addressed should the manuscript be published.

1. In Fig. 1C. the authors claim that the cells showed elevated G2/M ratio. This could be interpreted that the ratio of G2 to M cells has changed, which is not what the authors show. The authors should rephrase the sentence.

Response: Thank you for pointing out our inaccurate description. We have rephrased the sentence to accurately reflect our data. The original text, "...elevated G2/M ratios," has been changed to "...an increase in the proportion of cells in G2/M phase (Fig. 1C)." in the revised manuscript. We believe this new phrasing more precisely conveys that a larger fraction of the cell population is in the G2 or M phase of the cell cycle.

2. The results of the transcriptomics are not very informative. Why was this done? Also, the sentence "This paradox implies that chromosomal rearrangements primarily induce mitotic stress through physical constraints rather than transcriptional reprogramming" is very confusing. What transcriptional reprogramming have the authors in mind which would lead to whole genome doubling?

Response: We thank the reviewer for these insightful comments. The transcriptomic analysis was conducted to address a specific hypothesis. Prompted by observations that pathways like PI3K-Akt and MAPK can induce cell cycle alterations and promote

whole-genome doubling (Lacroix & Maddox, 2012), we sought to determine if similar transcriptomic changes explained the mitotic delay in our models. The negative result from the pathway enrichment analysis—showing no such consistent changes—is itself a critical finding, as it rules out this major mechanistic possibility.

We acknowledge that the term "transcriptional reprogramming" was confusing. It was intended to mean a consistent gene expression signature that could mechanistically explain the delay, which we did not find. We have removed the term "paradox" and revised the text for greater clarity, explicitly stating that the lack of pathway-level changes points to physical constraints. The relevant section has been rewritten, and enrichment analysis of three translocated cell lines were added in the revised manuscript and pasted in below:

Figure EV2. Gene Ontology enrichment analysis of differentially expressed genes.

Gene Ontology enrichment analysis of differentially expressed genes in chromosome-translocated diESCs (Chr1+2', Chr4+5, and Chr2+1) compared to WT

diESCs. The x-axis indicates the log₁₀-transformed FDR values, and the y-axis lists the enriched pathway terms. Red bars represent upregulated pathways, and blue bars represent downregulated pathways.

Relative descriptions:

“To determine if consistent transcriptional alterations underlie mitotic progression delays in chromosome-translocated cells, we performed transcriptomic analysis (Data ref: Wang *et al*, 2022). We identified 67 consistently dysregulated genes across all translocated cell lines (Fig. 1D). However, pathway enrichment analysis of these genes revealed no direct association with cell cycle checkpoints (Fig. 1E). Furthermore, we did not observe consistent enrichment patterns across the three individual cell lines (Fig. EV2). Collectively, these results suggest that the observed mitotic stress is likely induced by physical constraints rather than consistent transcriptomic changes.”

3. The authors claim that Fig. 2a "...demonstrate complete failure of chromosomal segregation at anaphase termination". This is not shown in the figure at all.

Response: We thank the reviewer for raising this important point. In response to his comment, we have now included 3D movie data in the revised manuscript to more clearly illustrate the failure of chromosomal segregation during anaphase termination (please see Movies EV1-5). These movies provide a comprehensive view of chromosome dynamics from multiple angles and unequivocally reveal the presence of persistent chromosome bridges in Chr2+3 and Chr2+1 diESCs at anaphase

termination, thereby confirming the failure of segregation.

4. In figure 2A, the authors claim that the bridges are due to the large chromosome, but they never show any evidence that the arising chromatin bridges are indeed the fusion chromosomes. This evidence has to be provided. Also, what fraction of cells shows chromatin anaphase bridges? Do all cells with bridges become tetraploid?

Response: We thank the reviewer for raising these key points. In response to his first question, we have added whole-chromosome DNA FISH and found that chromosome bridges perfectly merged with fusion chromosomes (please see Fig. EV3A, relative data and descriptions were pasted in below).

Fig. EV3A. FISH detection of lagging chromosomes at anaphase. Scale bar = 5 μ m.

Relative descriptions:

“To ascertain whether lagging chromosomes are fused chromosomes, we used whole chromosome FISH to label fused chromosomes, and found that the lagging chromosomes were perfectly merged with fish probes (Fig. EV3A).”

In response to his second and third questions, we have included fraction of cells with chromatin anaphase bridges and cell re-coalescence in our revised manuscript (please see revised Fig. 2B). Anaphase bridges formed in Chr4+5, Chr2+3 and Chr2+1 diESCs, but only a small fraction of cells with anaphase bridges in Chr2+3 and Chr2+1 diESCs became tetraploid. Reconstructed figure and descriptions were pasted in below:

Fig. 2B. Time-lapse imaging of metaphase-to-anaphase progression (0 min = metaphase plate formation). White dotted circles indicate cells undergoing mitosis, and white arrows mark lagging chromosomes. Scale bar = 5 μ m. For each group, statistics for the number of cells with anaphase bridges (AB), cell re-coalescence (CR) and total observed cells (TC) are provided.

Relative descriptions:

“Time-lapse quantification revealed 6.1% (8/131) Chr4+5 diESCs, 82.1% (64/78) Chr2+3 diESCs and 83.9% (94/112) Chr2+1 diESCs formed chromatin bridges at

anaphase, but cell re-coalescence was only observed in Chr 2+3 and Chr2+1 diESCs but not in Chr4+5 diESCs (Fig. 2B; Movies EV6-10).”

5. The authors claim that the chromatin bridges arising from fusion chromosomes lead to re-fusion of the chromosome masses into one tetraploid nucleus. This is highly unusual. Most of the time, chromatin bridges result in binucleated cells (e.g., Davoli, de Lange, 2012, Afonso et al, 2014). The authors cite the Afonso paper, and talk about cytokinesis failure, but this is not what they present in their figures.

Response: We thank the reviewer for these insightful comments. To better illustrate the process of tetraploid nucleus formation resulting from chromosome translocation, we have now included live-cell imaging movies of all chromosome translocated diESCs and WT diESCs as supplementary material (please see Movies EV6-10). In these movies, although binucleated cells do appear—consistent with the previous report (Davoli & de Lange, 2012)—chromatin bridges persistently connect the daughter nuclei throughout the process. This continuous connection suggests that binucleation represents an intermediate state preceding full tetraploid nucleus formation. Furthermore, through phalloidin staining of F-actin, we observed multiple nuclei (bi- and trinucleated cells) specifically in Chr2+3 and Chr2+1 diESCs, but not in other cell lines. Notably, chromatin bridges were consistently present in all such multinucleated cells (Fig. EV3b), reinforcing the idea that binucleation serves as a transitional stage in the re-coalescence process. Relevant figures and descriptions have been added to the manuscript and pasted in below:

Fig. EV3B. Immunofluorescence staining of Chr2+1 and Chr2+3 diESCs. Phalloidin staining indicates the spread of cytoskeleton, and white dotted circles label cells with binucleate and trinucleate. Scale bar = 5 μ m.

Relative descriptions:

“Previous studies have shown that chromatin bridges during telomere crisis can result in binucleated or multinucleated cells (Davoli & de Lange, 2012). Consistent with these findings, we frequently observed binucleated cells during the re-coalescence of Chr2+3 and Chr2+1 diESCs (Fig. 2B; Movies EV9 and EV10). Extending these previous reports, we found that chromatin bridges persisted not only in binucleated cells but also in tri-nucleated cells (Fig. EV3B). Taken together, our results suggest that binucleation likely represents an intermediate stage in the process of cell re-coalescence.”

We also thank the reviewer for pointing out the inappropriate citation of Afonso et al., 2014. Since that study describes lagging chromosomes and delayed nuclear envelope reformation—but not cytokinesis failure—we agree that it does not adequately support our original argument regarding cytokinesis impairment. We have

therefore replaced this reference with more appropriate ones in the revised manuscript (Lacroix & Maddox, 2012).

6. The title in Fig. 2B Ratio of cell fusion is completely misleading, because there was certainly no cell fusion (meaning fusion between two independent cells) occurring.

Response: We thank the reviewer for pointing out our inaccurate description. We have changed the term “cell fusion” with “cell re-coalescence” in manuscript and Fig. 2B and Fig. 3E. Here the term ‘re-coalescence’ refers to a phenomenon in which two nascent daughter cells fail to complete cytokinesis, retain a thin cytoplasmic bridge, and eventually rejoin to form a single cell. This is distinct from canonical cell fusion between independent cells.

7. How was the spindle length measured? There is no information in the methods, and none of the images shown in the figures would allow measuring spindle length.

Response: We thank the reviewer for this helpful suggestion. In response, we have now provided a detailed description of the spindle length measurement protocol in the Methods section. Additionally, as illustrated in the new panel Fig. 2C, spindle length was defined and measured as the distance between the two spindle poles at the end of anaphase. In this representative image of a Chr2+3 diESC, the yellow solid line indicates the measured spindle length, while the white dotted lines highlight the positions of the lagging chromosomes for context. Reconstructed figure and descriptions were pasted in below:

Fig. 2C. Measurements of spindle axis and the longest chromosomes. The yellow solid line indicates spindle axis, and white dotted lines indicate the longest chromosomes. Scale bar = 5 μm .

Relative descriptions:

“The lengths of the longest chromosomes and the spindles (pole-to-pole distance) were measured at the end of anaphase (Fig. 2C).”

8. Why do the tetraploid cells tolerate the long chromosomes? Do they have longer spindles? This has not been shown in the manuscript, and in fact, that are several papers suggesting that the spindle length does not scale with increasing ploidy (Storchova et al, 2006; Cohen-Sharir et al, 2021; Gudlin et al, 2025 and more). Did the length of spindle change from day 1 to day 26 in tetraploid ESCs (Fig. 1E)?

Response: We thank the reviewer for these critical questions and for referencing the key literature on this topic. We acknowledge that several studies in other systems have reported a lack of spindle scaling with ploidy. In our experimental system, however, we observed a clear positive correlation between ploidy and spindle length. Precise measurements in tetraploid ESCs (from WT, Chr2+3, and Chr2+1 lines), with haploid

and diploid WT ESCs as controls, confirm that spindle length increases with ploidy (revised Fig. 2F). Importantly, the Chr2+3 and Chr2+1 tetraploid ESCs, which harbor the exceptionally long fused chromosomes, developed even longer spindles than WT tetraploid cells. A key finding is that the average half-spindle length in these lines significantly exceeded the average length of the longest chromosomes (revised Fig. 2F; Fig. EV3C). We propose that this specific adaptation of elongated spindle architecture is a central mechanism enabling the tolerance and successful segregation of long chromosomes in our models.

Regarding temporal stability, our tracking of spindle length from early to late culture time points (up to day 26) revealed no significant change in Chr2+3 and Chr2+1 tetraploid ESCs (Fig. EV3D), demonstrating that the extended spindle phenotype is a stable feature.

The relevant figure panels and descriptions are provided below for your review:

Fig. 2F. Spindle axis length comparison between haploid, diploid and tetraploid cells. WT-1N: n=23; WT-2N: n=63; WT-4N: n=31; 2+3-4N: n=21; 2+1-4N: n=21. Each data point represents the measured length of the spindle axis. Statistical analyses were

performed using unpaired *t*-test. Data are presented as mean±SEM, with **** p value

of 1.2e-05, 2.4e-14, 8.52e-12 and 8.53e-12 rightward correspondingly.

Fig. EV3D. Comparison of spindle axis in tetraploid cells on day 1 and day 26. Day1 Chr2+3-4N: n=21; Day26 Chr2+3-4N: n=26; Day1 Chr2+1-4N: n=26; Day26 Chr2+1-4N: n=31. Each data point represents one spindle axis. Statistical analysis was performed using unpaired *t*-test. Data are presented as mean±SEM, with ns p value of 0.6598 and 0.3701 rightward correspondingly.

Relative descriptions:

“Tetraploid cells typically have larger volumes, which theoretically accommodates enlarged mitotic spindles. To test this, we quantified spindles in tetraploid ESCs from WT, Chr2+3, and Chr2+1 lines, with haploid and diploid WT ESCs as controls. As expected, spindle length increased with ploidy. Moreover, tetraploid Chr2+3 and Chr2+1 ESCs exhibited even longer spindles than WT tetraploids (Fig. 2F). In these two lines, the average half-spindle length significantly exceeded that of the longest chromosomes (Fig. 2F; Fig. EV3C). Consistent with this

adaptation, both Chr2+3 and Chr2+1 tetraploid ESCs maintained stable spindle dimensions over extended culture and achieved successful chromosome segregation in all observed anaphase events (102 and 105 events, respectively), despite frequent lagging chromosomes (Fig. 2G; Fig. EV3D).”

9. Fig. 3A shows example for Aurora B localization, but this is just one cell per sample, and no quantification can be used. This needs to be added. Also, the amount of Aurora B in spindles with and without bridges should be compared.

Response: We thank the reviewer for this helpful suggestion. Accordingly, we have performed Aurora B immunofluorescence analysis in WT and chromosome-translocated diESCs at the end of anaphase, and now provide a quantitative summary of the frequency of cells exhibiting chromosome bridges (please see updated descriptions in revised manuscript). Furthermore, we quantitatively compared the intensity of Aurora B staining in cells with and without chromosome bridges. This analysis revealed that cells with bridges possess significantly higher levels of Aurora B (please see revised Fig. 3A and 3B). The relevant figure panels and descriptions are provided below for your review:

Fig. 3. A. Aurora B localization (purple) and chromatin (blue) at the end of anaphase. Scale bar = 5 μ m. **B.** Quantification of Aurora B at the end of anaphase. WT: n=10; Chr1+2': n=10; Chr4+5-N-B (Chr4+5 without bridge): n=7; Chr4+5-B (Chr4+5 with bridge): n=8; Chr2+3: n=8; Chr2+1: n=8. Each data point shows the amount of Aurora B relative to that of chromatin. Statistical analysis was performed using unpaired *t*-test. Data are presented as mean \pm SEM, with a ns p value of 0.1088, ***p value of 2.06e-04, and ****p value of 4.5e-05, 4.6e-05, 4.6e-05 rightward correspondingly.

Relative descriptions:

“To investigate the mechanistic link between persistent lagging chromosomes and spindle elongation, we quantified the localization of Aurora B kinase. This key component of the chromosomal passenger complex monitors chromosome segregation and prevents premature de-condensation during anaphase (Afonso *et al*, 2014). Immunofluorescence quantification of Aurora B in late anaphase cells revealed the following incidences of chromosome bridges: WT (0/10), Chr1+2' (0/10), Chr4+5 (8/15), Chr2+3 (8/8), and Chr2+1 (8/8) diESCs. Notably, Chr4+5 diESCs with bridges, as well as Chr2+3 and Chr2+1 diESCs, exhibited heightened Aurora B accumulation at these sites (Fig. 3 A, B). This suggests that Aurora B may coordinate spatial adaptations in spindle architecture to ensure chromosomal resolution.”

10. Aurora B inhibition is causing a lot of phenotypes. In fact, polyploidy upon treatment with Aurora B inhibitor is among the most usual phenotypes. Why is this not happening here in the wild type (Fig. 3E)? Why are the cell cycle profiles so

different for Chr4+5 and Chr2+1 in Aurora B inhibition experiment (Fig. 3E)?

Response: We thank the reviewer for raising these important questions. Regarding the first point on polyploidy induction by Aurora B inhibition, we fully agree that this is a well-documented phenomenon. In our own data, a minor increase in polyploidy was indeed observed in WT diESCs upon Aurora B inhibitor treatment (please see revised Fig. 3F). To further validate this, we isolated and cultured the putative polyploid subpopulation from WT diESCs—with and without Aurora B inhibition. Only the inhibitor-treated group gave rise to a stably proliferating culture, which was confirmed by FACS and karyotyping to be polyploid (please see revised Fig. EV4C). These results are fully consistent with prior literature and confirm that Aurora B inhibition can indeed induce polyploidization in WT diESCs under permissive culture conditions (Hindriksen *et al*, 2015).

As for the second question concerning the differing cell cycle profiles between Chr4+5 and Chr2+1 diESCs under Aurora B inhibition, we appreciate the reviewer's careful observation. The original Fig. 3E presented ploidy data collected at different time points (23 days for Chr4+5 and 18 days for Chr2+1), which may have misleadingly suggested divergent responses. To enable a rigorous and temporally matched comparison, we have now re-analyzed the ploidy of all lines by FACS at a uniform 18-day time point and have updated Fig. 3F in the revised manuscript accordingly. The relevant text and figures have been incorporated into the revised manuscript as shown below:

Fig. 3F. Analysis of ploidy after Aurora B inhibition in WT, Chr4+5 and Chr2+1 diESCs. Blue rectangle: sorting of cells with DNA content equal to 2n. Pink rectangle: analyses of cells with DNA content > 4n.

Fig. EV4C. FACS and karyotyping analysis of tetraploid WT ESCs. Tetraploid cells were isolated from WT diESCs following 18 days of Aurora B inhibition. The pink rectangle highlights the population with a DNA content > 4n. Scale bar = 5 μm.

Relative descriptions:

“FACS profiling over 18 days confirmed robust progressive polyploidization in Chr4+5 diESCs (Fig. 3F). Aurora B inhibition markedly accelerated polyploidization in Chr2+1 diESCs, while resulting in only a minor increase in polyploidy in WT

diESCs—a finding consistent with the previous report (Fig. 3F; Fig. EV4C) (Hindriksen *et al*, 2015).”

11. The figure 3D, Chr4+5 is basically indiscernible, it is not clear what it shows.

Response: We thank the reviewer for this observation. We agree that the original Fig. 3D did not clearly illustrate the process of cell re-coalescence in Chr4+5 and Chr2+1 diESCs under Aurora B inhibition. To address this, we have replaced the image in Fig. 3D with a clearer micrograph and have outlined the nuclei with white dotted lines to better visualize the process (please see the revised Fig. 3E). Furthermore, we have provided corresponding time-lapse movies (please see Movies EV11-13) that comprehensively capture the entire re-coalescence process, offering additional dynamic evidence. The relevant figure panels and descriptions are provided below for your review:

Fig. 3E. Mitotic progression in Aurora B-inhibited WT, Chr4+5 and Chr2+1 diESCs.

White dotted circles indicate cells undergoing mitosis. For each group, statistics for the number of cell re-coalescence (CR) and total observed cells (TC) are provided.

Scale bar = 5 μ m.

12. What is "increased chromosome load"? (last paragraphs of chapter Aurora B kinase regulates spindle adaptation).

Response: Thank the reviewer for the helpful suggestion. We have changed “increased chromosome load” with “increased chromosome length” in the revised manuscript.

13. The discussion is highly speculative. The authors had a nice idea and show some interesting data, but the data is not complete and allow other, potentially more interesting interpretations.

Response: We thank the reviewer for this valuable suggestion. In the revised version, we have substantially refined the *Discussion* to reduce speculative statements and to strengthen mechanistic and experimental grounding. Specifically, we (1) reorganized the discussion to maintain a clear logic from mechanistic findings to evolutionary and physiological implications; (2) replaced uncertain conjectures with interpretations supported by *in vivo* evidence from our previous chromosome-fusion mouse lines, which directly validate the spindle-length constraint observed in this study; and (3) incorporated a deeper analysis of how spindle geometry may act as an intrinsic cellular limit influencing both karyotype evolution and polyploidization during development and disease.

These revisions ensure that all interpretations are presented as hypothesis-driven

extensions of our data rather than open speculation. The overall discussion now provides a more balanced and integrated narrative linking quantitative findings, evolutionary context, and physiological relevance. We sincerely appreciate the reviewer's insightful comment, which helped us substantially improve the clarity, focus, and scientific rigor of this section.

References

- Hindriksen S, Meppelink A, Lens SMA (2015) Functionality of the chromosomal passenger complex in cancer. *Biochem Soc T* 43: 23-32
- Lacroix B, Maddox AS (2012) Cytokinesis, ploidy and aneuploidy. *J Pathol* 226: 338-351

Reviewer #2:

The authors show here that chromosome length in mammalian cells is constrained by a biophysical limit governed by spindle geometry. Their findings address the structural basis for genome fragmentation in animals and provide a general mechanism linking chromosome size, spindle dynamics, and genome stability. The topic is of great importance and timely. However, there are a few concerns that should be addressed before its acceptance for publication.

1. The authors should annotate the phenotype of real-time imaging to visualize the chromosome damage using reports;

Response: Thank the reviewer for this helpful suggestion. To clearly show the process of cell re-coalescence, we have outlined the nuclei with white dotted lines to better visualize the process (please see the revised Fig. 2B, 3E). Furthermore, we have provided corresponding time-lapse movies (please see Movies EV6-13) that comprehensively capture the entire re-coalescence process, offering additional dynamic evidence. The relevant figure panels and descriptions are provided below for your review:

Fig. 2B. Time-lapse imaging of metaphase-to-anaphase progression (0 min = metaphase plate formation). White dotted circles indicate cells undergoing mitosis, and white arrows mark lagging chromosomes. Scale bar = 5 μ m. For each group, statistics for the number of cells with anaphase bridges (AB), cell re-coalescence (CR) and total observed cells (TC) are provided.

Fig. 3E. Mitotic progression in Aurora B-inhibited WT, Chr4+5 and Chr2+1 diESCs. White dotted circles indicate cells undergoing mitosis. For each group, statistics for the number of cell re-coalescence (CR) and total observed cells (TC) are provided. Scale bar = 5 μ m.

2. The authors argue for the length alternation and should perform unbiased quantification of telomere-telomere distance.

Response: We thank the reviewer for raising this important point regarding unbiased quantification. To directly address whether the lagging chromosomes correspond to the translocated ones, we performed whole-chromosome FISH, which confirmed that the lagging chromosomes precisely match the structurally rearranged chromosomes (please see the revised Fig. EV3A).

Furthermore, in the revised manuscript, we have included a clear explanation of our measurement strategy. As illustrated in the representative image in the revised Fig. 2C, the yellow solid line indicates the spindle length (pole-to-pole distance), while the white dotted lines mark the positions of the lagging chromosomes; the lengths of these lines correspond to the quantified distances. In addition, we now provide full quantitative data from multiple cells summarizing the measured lengths of lagging chromosomes as supplementary material (please see the revised Fig. EV3C). Together, these approaches provide a clear and measurable basis for our analysis of telomere-to-telomere distance and chromosome behavior during mitosis. The relevant figure panels and descriptions are provided below for your review.

Fig. EV3a. FISH detection of lagging chromosomes at anaphase. Scale bar = 5 μ m.

Relative descriptions:

“To ascertain whether lagging chromosomes are fused chromosomes, we used whole chromosome FISH to label fused chromosomes, and found that the lagging chromosomes were perfectly merged with fish probes (Fig. EV3A).”

Fig. 2C. Measurements of spindle axis and the longest chromosomes. The yellow solid line indicates spindle axis, and white dotted lines indicate the longest chromosomes. Scale bar = 5 μ m.

Relative descriptions:

“The lengths of the longest chromosomes and the spindle (pole-to-pole distance)

were measured at the end of anaphase (Fig. 2C).”

Fig. EV3C. Measurements of the longest chromosomes observed in live-cell imaging at the end of anaphase. WT: n=108; Chr1+2': n=44; Chr4+5: n=45; Chr2+3: n=58; Chr2+1: n=76.

Relative descriptions:

“In these two lines, the average half-spindle length significantly exceeded that of the longest chromosomes (Fig. 2F; Fig. EV3C).”

Reviewer #3:

I read the concise manuscript of Zhao and colleagues with great interest. Pioneering work of the group of Ingo Schubert has already shown more than 25 years ago that there seems to be an upper limit of chromosome size in plant cells. They found indications that the upper limit for a chromosome arm is half of spindle axis length. If arms are bigger mitotic failure occurs.

The authors of the current manuscript pioneered recently chromosome engineering in mice by inducing chromosome fusions using CRISPR/Cas. As mouse chromosomes are telocentric by cuts in the centromere-distal telomer and next to the centromere of another chromosome fusion can be obtained. Thus, depending on length of the fusion partners different sized chromosome were produced. The authors obtained these set of fusions in ESCs. They found an arm length of about 308 Mb as maximum tolerated length for diploid cells. Interestingly in tetraploid cells longer arm sized were tolerated due to the fact that spindle axis is also larger in these cells. These findings nicely sustain Ingo Schubert's hypothesis and demonstrate that show that the limitation is not restricted to cells with a rigid cell wall. This is a very nice and well-done study and I do not have any suggestions to improve it.

Response:

We sincerely thank the reviewer for the thoughtful and encouraging evaluation of our work. We deeply appreciate the recognition of the conceptual link between our

findings and the pioneering studies by Ingo Schubert on chromosome-size limits in plant cells. We are pleased that the reviewer found our results to support and extend this long-standing hypothesis to mammalian systems. We are grateful for the reviewer's positive assessment and for taking the time to review our manuscript.

11th Dec 2025

Manuscript Number: MSB-2025-13173R

Title: Chromosome length is constrained by spindle scaling to ensure faithful mitosis in mammals

Dear Dr Wang,

Thank you for the submission of your revised manuscript to Molecular Systems Biology. I am pleased to inform you that we will be able to accept your manuscript pending the following final amendments and appropriate response to reviewers:

1) In the main manuscript file, please move the author list and affiliations to the title page.

2) Please include keywords to max. 5.

3) Please format the Data availability section according to the example below.

"The datasets and computer code produced in this study are available in the following databases:

- Chip-Seq data: Gene Expression Omnibus GSE46748 (<https://www.ncbi.nlm.nih.gov/geo/query/acc.cgi?acc=GSE46748>)

- Modeling computer scripts: GitHub (<https://github.com/SysBioChalmers/GECKO/releases/tag/v1.0>)

- [data type]: [full name of the resource] [accession number/identifier] ([doi or URL or identifiers.org/DATABASE:ACCESSION])"

4) Please rename "Conflict of Interest" to "Disclosure and competing interests statement". We updated our journal's competing interests policy in January 2022 and request authors to consider both actual and perceived competing interests. Please review the policy <https://link.springer.com/partners/embo-press/editorial-policies#Competing%20interest%20disclosures> and update your competing interests if necessary.

5) In the Methods, please take care of the following:

- The Materials and Methods section should be renamed to "Methods".

- Cell lines: Please also be sure to include a sentence in the Methods as to whether or not the cell lines were recently authenticated and tested for mycoplasma contamination. Please also be sure to update the Author Checklist with this information and where it can be found in the manuscript.

- Please ensure that a statement on whether or not blinding was done is included in the Methods even if no blinding was done.

Please also be sure to update the Author Checklist with this information and where it can be found in the manuscript.

6) Please place individual sections of the manuscript in the following order: Title page - Abstract & Keywords - Introduction - Results - Discussion - Methods - Data Availability - Acknowledgements - Disclosure and Competing Interests Statement - References - Figure Legends - Expanded View Figure Legends.

7) For the figures and figure legends, please take care of the following:

- Please indicate the statistical test used for data analysis in the legends of figures 1E, EV2

- Please note that the error bars are not defined in the legends of figures 1C, 4B, EV4 B

8) Please remove the legends for the movies from the manuscript; each legend needs to have its own text file and then each movie should be zipped up with its corresponding legend so that there are 13 individual zip folders: Movie EV1-Movie EV13.

9) The synopsis image does not currently meet our requirements for size - while we can resize the PNG ourselves, when sized to the required 550 pixels wide, the graphic is too tall, as it must be 300-600 pixels high. Please adjust the figure.

10) Source Data should be organized as a single source data file (zipped) per figure for main figures, not in one folder for all Source Data, e.g. all the Source data files for figure 1 need to be saved in a single folder and this needs to be zipped and then uploaded as "SD figure 1.zip" file. The Source Data checklist should be uploaded separately as a Related Manuscript File.

11) As part of the EMBO Publications transparent editorial process initiative (see our policy here:

https://www.embopress.org/transparent-process#Review_Process), Molecular Systems Biology will publish online a Peer Review File (PRF) to accompany accepted manuscripts. This file will be published in conjunction with your paper and will include the anonymous referee reports, your point-by-point response and all pertinent correspondence relating to the manuscript. Let us know whether you agree with the publication of the PRF and as here, if you want to remove or not any figures from it prior to publication. Please note that the Authors checklist will be published at the end of the PRF.

12) After your paper is published, we may promote it on social media. If you have any handles or hashtags for Bluesky you would like included, please let us know.

13) Please provide a point-by-point letter INCLUDING my comments as well as the reviewer's reports and your detailed responses (as Word file).

I look forward to reading a new revised version of your manuscript as soon as possible.

Yours sincerely,

Poonam Bheda, PhD
Scientific Editor
Molecular Systems Biology

Reviewer #1:

The authors of this manuscript did a very good job in addressing the comments. The new version is significantly improved. I still have a few points which should be adjusted for publication.

1. Authors explained how they did the spindle-length measurements. Yet, without staining the spindle poles or the spindle itself, the measurements they perform are only an approximately of the pole-to-pole distance. Authors should clearly state that and should not name it pole-to-pole distance, neither spindle length. In their figure, they use the expression "length of spindle axes:", which is a very good description of what they really measure. They should use it also in the text.

2. Authors clarified that not all cells with a bridge become binucleated, and show the numbers. The sentence below is difficult to understand and should be formulated better.

Time-lapse quantification revealed 6.1% (8/131) Chr4+5 diESCs, 82.1% (64/78) Chr2+3 diESCs and 83.9% (94/112) Chr2+1 diESCs formed chromatin bridges at anaphase, but cell co-coalescence was only observed in Chr 2+3 and Chr2+1 diESCs but not in Chr4+5 diESCs (Fig. 2B; Movies EV6-10).

Figure 2B legend statistics: statistics for the number of cells with anaphase bridges (AB), cell re-coalescence (CR) and total observed cells (TC) are provided.

But they show not statistics, only the numbers. Has to be corrected. It is also suggesting that this is just one biological replicate. Or was there more independent experiments done? Would be good to mention.

3. Interestingly, the Chr.4+5 cells never show CR. This is at odds with the finding in Fig. 3F, where they show a significant fraction of polyploidy also in Chr.4+5 cells without Aurora B inhibition. The authors should explain this.

Reviewer #2:

The revised manuscript has addressed my concerns raised in the last submission. I vote the acceptance of revised manuscript.

Reviewer #3:

The authors did great job taking the suggestions of the reviewers into account to improve the manuscript. This is now a really nice and important piece of work that should be published without further ado.

Point-by-point response

We sincerely thank the Editor for the helpful guidance and all reviewers for their careful evaluation and constructive suggestions. In the following, we addressed the Editor's comments first, followed by point-by-point responses to the comments from each reviewer.

Editor:

Thank you for the submission of your revised manuscript to Molecular Systems Biology. I am pleased to inform you that we will be able to accept your manuscript pending the following final amendments and appropriate response to reviewers:

1) In the main manuscript file, please move the author list and affiliations to the title page.

Response: Thank you for the suggestion. The author list and affiliations have been moved to the title page in the revised manuscript.

2) Please include keywords to max. 5.

Response: Thank you for the suggestion. Five keywords have been included in the revised manuscript.

Relative descriptions:

Keywords: Mitosis; Chromosomal translocations; Spindle; Chromosome

length; polyploidization”

3) Please format the Data availability section according to the example below.

"The datasets and computer code produced in this study are available in the following databases:

- Chip-Seq data: Gene Expression Omnibus GSE46748

(<https://www.ncbi.nlm.nih.gov/geo/query/acc.cgi?acc=GSE46748>)

- Modeling computer scripts: GitHub

(<https://github.com/SysBioChalmers/GECKO/releases/tag/v1.0>)

- [data type]: [full name of the resource] [accession number/identifier] ([doi or URL or identifiers.org/DATABASE:ACCESSION])"

Response: Thank you for the guidance. We have reformatted the Data Availability section according to the provided example, listing the datasets and resources in the requested format with full database names, accession numbers, and corresponding URLs or identifiers.

Relative descriptions:

“The datasets produced in this study are available in the following databases:

- RNA-seq data: Genome Sequence Archive CRA030371

(<https://ngdc.cncb.ac.cn/gsa/search?searchTerm=CRA030371>)”

4) Please rename "Conflict of Interest" to "Disclosure and competing interests statement".

We updated our journal's competing interests policy in January 2022 and request authors to consider both actual and perceived competing interests. Please review the policy <https://link.springer.com/partners/embo-press/editorial-policies#Competing%20interest%20disclosures> and update your competing interests if necessary.

Response: We thank the editor for raising this important point. We have renamed "Conflict of Interest" to "Disclosure and competing interests statement." We have reviewed the updated competing interests policy and confirm that there are no actual or perceived competing interests to disclose.

5) In the Methods, please take care of the following:

- The Materials and Methods section should be renamed to "Methods".
- Cell lines: Please also be sure to include a sentence in the Methods as to whether or not the cell lines were recently authenticated and tested for mycoplasma contamination. Please also be sure to update the Author Checklist with this information and where it can be found in the manuscript.
- Please ensure that a statement on whether or not blinding was done is included in the Methods even if no blinding was done. Please also be sure to update the Author Checklist with this information and where it can be found in the manuscript.

Response: We thank the editor for raising these important points regarding the "Methods" section. The "Materials and Methods" section has been renamed to

“Methods.” We have added a statement in the Methods describing cell line authentication and mycoplasma testing, and included a statement indicating that no blinding was performed. The Author Checklist has been updated to reflect this information and to indicate where it can be found in the manuscript.

Relative descriptions:

“All cell lines were authenticated and routinely tested negative for mycoplasma contamination.”

“No blinding was performed.”

6) Please place individual sections of the manuscript in the following order: Title page - Abstract & Keywords - Introduction - Results - Discussion - Methods - Data Availability - Acknowledgements - Disclosure and Competing Interests Statement - References - Figure Legends - Expanded View Figure Legends.

Response: Thank you for the suggestion. We have reorganized the manuscript to follow the required section order: Title page, Abstract & Keywords, Introduction, Results, Discussion, Methods, Data Availability, Acknowledgements, Disclosure and Competing Interests Statement, References, Figure Legends, and Expanded View Figure Legends.

7) For the figures and figure legends, please take care of the following:

- Please indicate the statistical test used for data analysis in the legends of figures 1E,

EV2

- Please note that the error bars are not defined in the legends of figures 1C, 4B, EV4

B

Response: Thank you for pointing this out. We have revised the figure legends accordingly. The statistical tests used for data analysis are now specified in the legends of Figures 1E and EV2, and the definitions of the error bars have been added to the legends of Figures 1C, 4B, and EV4B.

Relative descriptions:

“Figure 1E. Gene Ontology (GO) enrichment analysis of 67 conserved DEGs across all translocation lines. Enrichment analysis was calculated by using Fisher’s exact test.”

“Figure EV2. Gene Ontology enrichment analysis of differentially expressed genes in chromosome-translocated diESCs (Chr1+2', Chr4+5, and Chr2+1) compared to WT diESCs. The x-axis indicates the log₁₀-transformed FDR values, and the y-axis lists the enriched pathway terms. Red bars represent upregulated pathways, and blue bars represent downregulated pathways. Enrichment analysis was calculated by using Fisher’s exact test.”

“Figure 1C. Cell cycle analysis of WT and chromosome-translocated diESCs. Each bar represents the cell cycle of three biological replicates (n=3). Data are presented as mean ± SD.”

“Figure 4B. Teratoma weights. Each bar represents the tumor weights of three

biological replicates (n=3). Data are presented as mean \pm SD.”

“Figure EV4B. Length comparison of the longest chromosomes in WT, Chr4+5 and Chr2+1 diESCs with and without Aurora B inhibitor. WT + Vehicle: n=59; WT + Aurora B inhibitor: n=58; Chr4+5 + Vehicle: n=57; Chr4+5 + Aurora B inhibitor: n=59; Chr2+1 + Vehicle: n=59; Chr2+1 + Aurora B inhibitor: n=27. Data are presented as mean \pm SD.”

8) Please remove the legends for the movies from the manuscript; each legend needs to have its own text file and then each movie should be zipped up with its corresponding legend so that there are 13 individual zip folders: Movie EV1-Movie EV13.

Response: Thank you for the instruction. We have removed all movie legends from the manuscript. Each movie legend has been saved as an individual text file, and each movie has been zipped together with its corresponding legend. A total of 13 zip files (Movie EV1–Movie EV13) have been prepared and uploaded accordingly.

9) The synopsis image does not currently meet our requirements for size - while we can resize the PNG ourselves, when sized to the required 550 pixels wide, the graphic is too tall, as it must be 300-600 pixels high. Please adjust the figure.

Response: Thank you for pointing this out. We have adjusted the synopsis image to meet the required dimensions. The revised PNG is now 550 pixels wide, with a height within the 300-600 pixel range, and has been uploaded with the revised submission.

10) Source Data should be organized as a single source data file (zipped) per figure for main figures, not in one folder for all Source Data, e.g. all the Source data files for figure 1 need to be saved in a single folder and this needs to be zipped and then uploaded as "SD figure 1.zip" file. The Source Data checklist should be uploaded separately as a Related Manuscript File.

Response: Thank you for the clarification. We have reorganized the Source Data so that all source data files for each main figure are collected into a single folder, which has been zipped and uploaded as an individual file ("SD figure 1.zip-SD figure 4.zip"). The Source Data checklist has been uploaded separately as a Related Manuscript File, as requested.

11) As part of the EMBO Publications transparent editorial process initiative (see our policy here: https://www.embopress.org/transparent-process#Review_Process), Molecular Systems Biology will publish online a Peer Review File (PRF) to accompany accepted manuscripts. This file will be published in conjunction with your paper and will include the anonymous referee reports, your point-by-point response and all pertinent correspondence relating to the manuscript. Let us know whether you agree with the publication of the PRF and as here, if you want to remove or not any figures from it prior to publication. Please note that the Authors checklist will be published at the end of the PRF.

Response: We agree with the publication of the Peer Review File (PRF) in

conjunction with the accepted manuscript and do not request the removal of any figures from the PRF.

12) After your paper is published, we may promote it on social media. If you have any handles or hashtags for Bluesky you would like included, please let us know.

Response: We are happy for the paper to be promoted on social media. At this time, we do not have any Bluesky handles or hashtags to include.

13) Please provide a point-by-point letter INCLUDING my comments as well as the reviewer's reports and your detailed responses (as Word file).

Response: We thank the Editor and the reviewers for their constructive and insightful comments. In response, we have prepared a detailed point-by-point response letter, provided as a Word file, which includes the Editor's comments, the full reviewer reports, and our corresponding responses to each point.

Reviewer #1:

The authors of this manuscript did a very good job in addressing the comments. The new version is significantly improved. I still have a few points which should be adjusted for publication.

1. Authors explained how they did the spindle-length measurements. Yet, without staining the spindle poles or the spindle itself, the measurements they perform are

only an approximately of the pole-to-pole distance. Authors should clearly state that and should not name it pole-to-pole distance, neither spindle length. In their figure, they use the expression "length of spindle axes:", which is a very good description of what they really measure. They should use it also in the text.

Response: Thank the reviewer for the helpful suggestion. We have changed the terms “pole-to-pole distance” and “spindle length” with “length of the spindle axis” or “spindle axis length” in the revised manuscript.

2. Authors clarified that not all cells with a bridge become binucleated, and show the numbers. The sentence below is difficult to understand and should be formulated better.

Time-lapse quantification revealed 6.1% (8/131) Chr4+5 diESCs, 82.1% (64/78) Chr2+3 diESCs and 83.9% (94/112) Chr2+1 diESCs formed chromatin bridges at anaphase, but cell co-coalescence was only observed in Chr 2+3 and Chr2+1 diESCs but not in Chr4+5 diESCs (Fig. 2B; Movies EV6-10).

Figure 2B legend statistics: statistics for the number of cells with anaphase bridges (AB), cell re-coalescence (CR) and total observed cells (TC) are provided.

But they show not statistics, only the numbers. Has to be corrected. It is also suggesting that this is just one biological replicate. Or was there more independent experiments done? Would be good to mention.

Response: We thank the reviewer for pointing this out. First, we agree that the original sentence describing the frequencies of chromatin bridge formation and cell

co-coalescence was difficult to follow. We have therefore rewritten this sentence in the main text to improve clarity and to more explicitly distinguish between the occurrence of anaphase bridges and the subsequent cell co-coalescence outcome.

Second, regarding the data shown in Figure 2B, we would like to clarify that the number reported for anaphase bridges (AB), cell re-coalescence (CR), and total observed cells (TC) represent absolute cell counts, rather than statistical test results. These numbers were obtained by manually scoring all cells that underwent complete mitosis during live-cell imaging experiments. Importantly, these counts were pooled from multiple independent live-cell imaging experiments, rather than derived from a single biological replicate. We have now clarified this point in the legends of 2B, 3E to avoid confusion.

Relative descriptions:

“Time-lapse imaging showed that chromatin bridges at anaphase occurred in 6.1% (8/131) of Chr4+5 diESCs, 82.1% (64/78) of Chr2+3 diESCs, and 83.9% (94/112) of Chr2+1 diESCs. However, the presence of a chromatin bridge did not necessarily result in cell co-coalescence: co-coalescence was observed in Chr2+3 and Chr2+1 diESCs but not in Chr4+5 diESCs (Fig. 2B; Movies EV6-10).”

“Figure 2B. For each group, numbers for AB, CR, and TC represent pooled cell counts from multiple independent live-cell imaging experiments, including cells with complete mitotic progression.”

“Figure 3E. For each group, numbers for cell re-coalescence (CR) and total observed cells (TC) represent pooled cell counts from multiple independent live-cell imaging experiments, including cells with complete mitotic progression.”

3. Interestingly, the Chr.4+5 cells never show CR. This is at odds with the finding in Fig. 3F, where they show a significant fraction of polyploidy also in Chr.4+5 cells without Aurora B inhibition. The authors should explain this.

Response: We thank the reviewer for this comment. The discrepancy can be explained by the intrinsic background noise of flow cytometry-based DNA content analysis in combination with altered cell-cycle distributions. As shown in the flow cytometry profiles of WT ESCs in Fig. 1B and Fig. 3F, a small population of cells with DNA content greater than $4n$ is consistently detected even under normal conditions, indicating a baseline level of noise in the assay. This background signal is therefore not unique to Chr4+5 cells. Importantly, as demonstrated in Fig. 1B and Fig. 1C, all rearranged diESCs, including Chr4+5 diESCs, display a marked shift in cell-cycle distribution, with an increased proportion of cells in the G2/M phase. This shift leads to an overall increase in the number of cells occupying the $4n$ DNA content region. As a consequence, the intrinsic background noise of the flow cytometry assay has a proportionally larger impact, resulting in an apparent increase in cells scored as $>4n$. Therefore, although Chr4+5 cells do not exhibit detectable cell re-coalescence (CR) by direct observation, their altered cell-cycle composition amplifies the contribution of flow cytometry background noise in the $4n$ and $>4n$ regions. We thus

interpret the elevated polyploid fraction observed in Fig. 3F as a technical overestimation rather than evidence of bona fide CR events in Chr4+5 diESCs.

Reviewer #2:

The revised manuscript has addressed my concerns raised in the last submission. I vote the acceptance of revised manuscript.

Response: We sincerely thank the reviewer for their careful re-evaluation of the revised manuscript and for acknowledging that the concerns raised in the previous round have been addressed. We appreciate the reviewer's time and support.

Reviewer #3:

The authors did great job taking the suggestions of the reviewers into account to improve the manuscript. This is now a really nice and important piece of work that should be published without further ado.

Response: We are grateful to the reviewer for the careful assessment of the revised manuscript and for the very positive evaluation of our work. We appreciate the reviewer's time and support.

2nd Jan 2026

Manuscript number: MSB-2025-13173RR

Title: Chromosome length is constrained by spindle scaling to ensure faithful mitosis in mammals

Dear Dr Wang,

Congratulations on an excellent manuscript, I am pleased to inform you that your manuscript has been accepted for publication in Molecular Systems Biology. Thank you for your comprehensive response to referee concerns. It has been a pleasure to work with you to get this to the acceptance stage.

You may qualify for financial assistance for your publication charges - either via a Springer Nature fully open access agreement or an EMBO initiative. Check your eligibility: <https://link.springer.com/journal/44320/how-to-publish-with-us>

Yours sincerely,

Poonam Bheda, PhD
Scientific Editor
Molecular Systems Biology

>>> Please note that it is Molecular Systems Biology policy for the transcript of the editorial process (containing referee reports and your response letter) to be published as an online supplement to each paper. If you do NOT want this, you will need to inform the Editorial Office via email immediately. More information is available here: <https://link.springer.com/partners/embo-press/editorial-policies#Peer%20review>